# Complex Numerical Reasoning with Numerical Semantic Pre-training Framework

## Abstract

Multi-hop complex reasoning over incomplete knowledge graphs has been extensively studied, but research on numerical knowledge graphs remains relatively limited. Recent approaches focus on separately encoding entities and numerical values, using neural networks to process query encodings for reasoning. However, in complex multi-hop reasoning tasks, numerical values are not merely symbols; they carry specific semantics and logical relationships that must be accurately represented. Directly encoding numerical values often leads to the loss of such semantic information. In this work, we propose a Complex Numerical Reasoning with Numerical Semantic Pre-Training Framework **(CNR-NST)**. Specifically, we designed a joint link predictor to learn numerical semantics. The proposed framework is the first to enable binary operations on numerical attributes in numerical knowledge graphs, allowing new numerical attributes to be inferred from existing knowledge. The CNR-NST framework can perform binary operations on numerical attributes in numerical knowledge graphs, enabling it to infer new numerical attributes from existing knowledge. Our approach effectively handles up to 102 types of complex numerical reasoning queries. On three public datasets, CNR-NST demonstrates SOTA performance in complex numerical queries, achieving an average improvement of over 40% compared to existing methods. Notably, this work expands the range of query types for complex multi-hop numerical reasoning and introduces a new evaluation metric for numerical answers, which has been validated through comprehensive experiments.

## 1 Introduction

Complex Query Answering (CQA) refers to the process of reasoning and performing computations on knowledge graphs by combining multiple entities and relationships to retrieve entities that fulfill specific logical conditions (Kotnis & García-Durán (2018)). This field has seen significant advancements, with research increasingly focused on enhancing the accuracy of models in handling intricate query tasks.(Zhu et al. (2022)) (Arakelyan et al. (2020)) Despite this progress, real-world knowledge graphs are not limited to discrete entity-relation knowledge; they also contain numerous numerical attributes, such as birth dates, event times, and territorial sizes of countries. Numerical knowledge graphs, therefore, offer a more nuanced approach to modeling real-world query (Xue et al. (2022)). Figure 1 presents an example of the FB15K numerical knowledge graph (Kotnis & García-Durán (2018)) illustrating three distinct query types: **Complex Queries Involving Only Entities**, such as $Q1$: Which film directors are married?; **Complex Numerical Queries**, such as $Q2$: Who are the married individuals born in August 1955?; and **Complex Numerical Queries with Calculations**, such as $Q3$:What is the combined population of Schleswig-Holstein and Dakar?

Traditional CQA methods are effective for multi-hop queries but encounter challenges in accurately capturing the subtle nuances of numerical semantics (Ren et al. (2023)). These methods neglect the inherent meanings of numerical attributes. Existing approaches, such as the Numerical Reasoning Network (NRN) (Bai et al. (2023a)), present a framework that encodes entities and numerical values separately. However, when using Sinusoidal (Sundararaman et al. (2020)) and DICE (Vaswani (2017)) encoding methods for numerical values, this approach encounters issues with sparsity in the knowledge graph. Many encoding regions lack corresponding numerical mappings in the knowledge graph, leading to incomplete learning of numerical semantics.

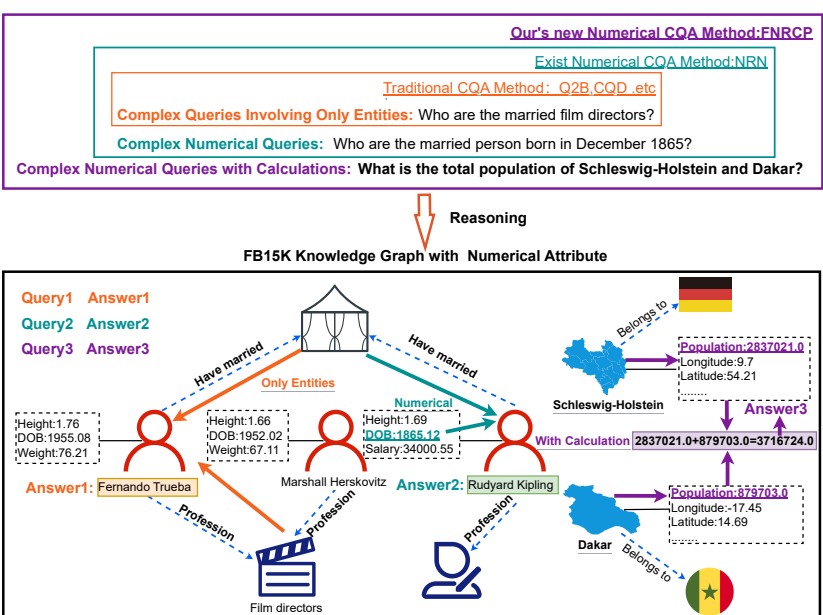

Figure 1: There are three types of queries: $Q1$ refers to Complex Queries Involving Only Entities, which involve complex reasoning tasks that solely include entities. $Q2$ refers to Complex Numerical Queries, which combine both numerical values and entities for reasoning. $Q3$ refers to Complex Numerical Queries with Calculations, where the query requires performing computations between two numerical values within the knowledge graph.

Handling complex numerical query tasks presents several significant challenges. First, current methods are constrained by their reasoning answers, which are limited to the numerical knowledge graph itself and cannot compute or infer new numerical answers from multiple values (like $Q3$). The ability to infer new numerical attributes from existing values in knowledge graphs is essential for practical applications. Second, unlike entities, numerical values are inherently continuous (Ren et al. (2023)), with their semantics influenced by factors such as units, ranges, and precision, establishing an intrinsic connection with their context (Kim et al. (2023)). Moreover, the sparsity of numerical data within knowledge graphs introduces additional challenges (Li et al. (2022)) that must be addressed to effectively tackle complex queries.

To tackle the challenges in complex numerical reasoning, we propose the CNR-NST framework:

**Numerical Binary Operation Operator.** As illustrated in Figure 2a, CNR-NST is capable of handling queries that involve numerical answers not present in the original knowledge graph, introducing new numerical query operators. Utilizing relevant theorems from fuzzy mathematics, we can perform operations on two fuzzy numbers within the real number domain. By representing numerical values as fuzzy sets, these operations can be effectively mapped to mathematical calculations between two numerical attributes within the knowledge graph (Detailed mathematical proof can be found in Appendix A.1).

**Numerical Semantic Learning.** To effectively capture the semantics of numerical values, we utilize a joint predictor to learn the relationships between entity attributes and their corresponding numerical values, thus facilitating knowledge transfer across different tasks. Figure 2b provides a comprehensive overview of this architecture.

**Complex Numerical Reasoning and Computation.** CNR-NST utilizes fuzzy sets to represent numerical values and entities, thereby avoiding the training of neural operators in an unrestricted numerical embedding space (as illustrated in Figure 2c). Fuzzy sets not only capture the uncertainty of numerical values during complex reasoning but also effectively represent the inherent fuzzy relationships within numerical data. During the reasoning process, the values of intermediate variables reflect the probability scores of corresponding entities or numerical values, thereby greatly enhancing the accuracy of numerical inference and reasoning.

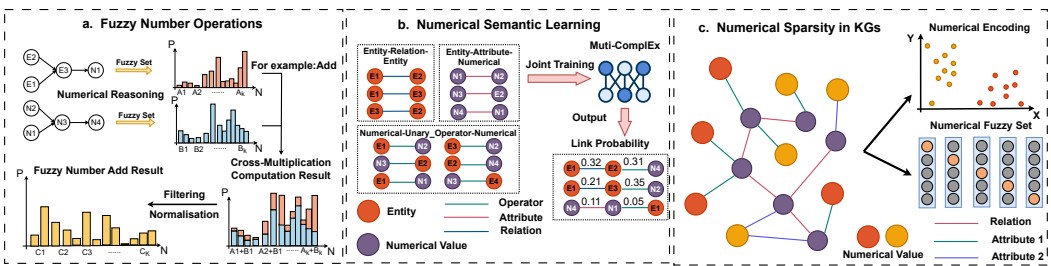

Figure 2: Figure a: The process of performing operations on two fuzzy numbers derived from reasoning. Figure b: The pre-training process for learning numerical semantics in CNR-NST. Figure c: Numerical sparsity in the knowledge graph and two numerical representation methods: direct encoding and fuzzy sets.

In summary, our contributions are:

- We introduced the **Binary Operation** operator for the first time in complex numerical queries, defining new operators and query types. The numerical binary operation operator enables the inference and computation of new attribute values from existing knowledge graphs, thereby enriching attribute knowledge and contributing to the construction of a more comprehensive knowledge graph. Additionally, we established more appropriate evaluation metrics for numerical answers.

- We introduced the CNR-NST framework, which learns various entity-numerical relationships during the pre-training stage. Our numerical semantic pre-training model can capture subtle nuances in numerical semantics (e.g., distinguishing between 180 cm in height and 180 kg in weight). By learning complex relationships and contextual information within the knowledge graph, the model achieves more accurate embedding and link prediction for numerical values across different attributes.

- CNR-NST can accurately represent the highly sparse numerical attributes in numerical knowledge graphs (Detailed statistical information on the numerical sparsity of the three public datasets can be found in Appendix B.1), significantly enhancing the accuracy of numerical predictions in query processing. Compared to the current state-of-the-art numerical reasoning models, CNR-NST achieves substantial performance gains across all 102 types of complex numerical queries.

## 2 PRELIMINARIES

**Knowledge Graph With Numerical Attributes.** Knowledge Graph $\mathcal{G} = (\mathcal{V}, \mathcal{R}, \varepsilon)$ contains the set of all entities $\mathcal{V}$ and the set of all relations $\mathcal{R}$. Each triplet $(h, r, t) \in \varepsilon$ is the set of facts in the knowledge graph. And Numerical Knowledge Graph $\mathcal{G}_N = (\mathcal{V}, \mathcal{R}, \mathcal{A}, \mathcal{N}, \varepsilon)$ not only contains entities $V$ and relations $\mathcal{R}$, but also contains the set of all attributes $\mathcal{A}$ and the set of all attribute values $\mathcal{N}$ (Pai & Costabello (2021)). Meanwhile, the triplet $(h, r, t)$ in the Numerical Knowledge Graph is defined as $\mathcal{G}_N = \{(h, r, t)\} \subset (\mathcal{V} \times \mathcal{R} \times \mathcal{V}) \cup (\mathcal{V} \times \mathcal{A} \times \mathbb{N})$, where $\mathcal{A} \cap \mathcal{R} = \varnothing$.

**Numerical FOL Query.** A complex numerical query is defined in existential positive first-order logic form, which can be recursively defined as:

1. Atomic Formulas: If $t_1, t_2, \ldots, t_n$ are variables or constants, and $P$ is an n-ary predicate, then $P(t_1, t_2, \ldots, t_n)$ is an atomic formula.

2. Compound formulas: It can be constructed using logical connectives $\wedge$ (and), $\vee$ (or), and quantifiers $\forall$ (for all) and $\exists$ (there exists).

**Numerical Complex Query Answering.** As every logic query can be converted into a disjunctive normal form, the complex query Q on a knowledge graph with numeric literals ($\mathcal{G}_N$) can be defined as:

$$q[X_?] = X_?.V_1, \ldots, V_i \in \mathcal{V}, N_1, \ldots, N_j \in \mathcal{N} : c_1 \vee c_2 \vee \cdots \vee c_n \tag{1}$$

$$c_i = e_{i,1} \wedge e_{i,2} \wedge \cdots \wedge e_{i,m}$$

Here, $c_i$ represents a conjunction of several atomic logical expressions $e_{i,j}$, where each $e_{i,j}$ can be one of the following expressions:

$$e_{i,j} = r\left(E, E'\right), \text{ with } E, E' \in \{V_1, \dots, V_i\}, E \neq E', r \in \mathcal{R}$$

$$e_{i,j} = a\left(E, C\right), \text{ with } E \in \{V_1, \dots, V_i\}, C \in \{N_1, \dots, N_j\}, a \in \mathcal{A}$$

$$e_{i,j} = f\left(C, C'\right), \text{ with } C, C' \in \{N_1, \dots, N_j\}, C \neq C', f \in \{\leq, \geq, =, \dots\}$$

$$e_{i,j} = b_f\left(C, C', N_{\text{result}}\right), \text{ with } C, C' \in \{N_1, \dots, N_j\}, N \text{ is any number}, b_f \in \{+, -, \times, \div, \dots\}$$

In the above equation, the variable $E$ represents a subset of entity $\mathcal{V}$, and the variable $C$ represents a subset of the numerical attribute set $\mathbb{N}$. The binary function $r_i$ determines whether a class $i$ relationship exists between two entities. The function $a_j$ determines whether an entity possesses a value for attribute $j$. The function $f$ checks whether a filtering condition, such as greater than or less than, is satisfied between two values. The function $b_f$ determines whether a quadratic operation between the first two values yields an answer.

The objective of complex reasoning is to find a valid assignment for the variables such that the query $q\left[X_?\right]$ holds true. Due to the incompleteness of the knowledge graph, uncertainty is introduced, meaning that $e_{i,j}$ is no longer a binary variable; instead, it represents the likelihood that the correspondence holds, with a generalized truth value ranging between [0,1]. For this reason, as follows (Arakelyan et al. (2020)), we formalise Eq.(1) as an optimisation problem:

$$q\left[X_?\right] = X_?.V_1, \dots, V_i \in \mathcal{V}, N_1, \dots, N_j \in \mathcal{N} = \arg\max\left(e_{1,1}\top \dots \top e_{1,m}\right)\bot \dots \\ \bot\left(e_{n,1}\top \dots \top e_{n,m}\right) \tag{2}$$

where $e_{n,m}$ is the probability score inferred by Multi-ComplEx based on the corresponding atomic formula. $\bot$ and $\top$ are generalisations of fuzzy logic over [0,1] for conjunctive and disjunctive extraction, and we chose the product t-norm and t- connorm (Hájek (2013)) as natural connectives in fuzzy logic in this paper.

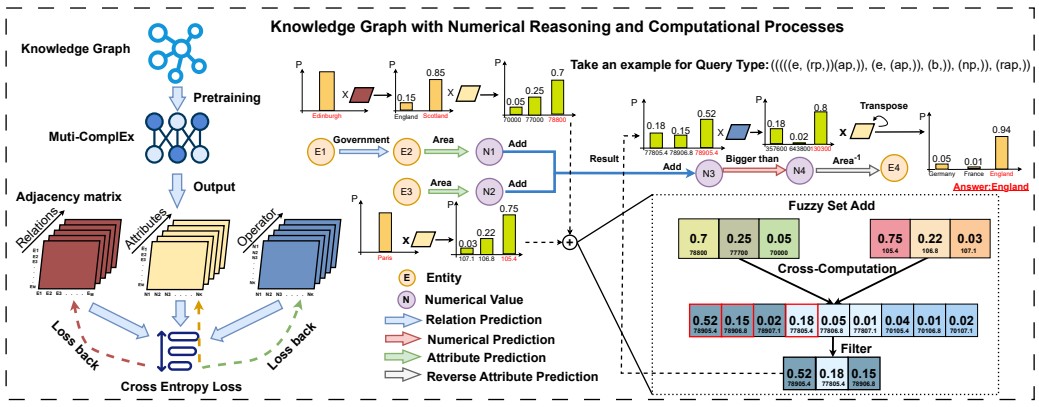

Figure 3: CNR-NST for Numerical Reasoning and Computation. For instance, in the case of a complex multi-hop query, the intermediate variable at each step is quantified by a score. The score for the subsequent hop is then computed using a pre-trained Knowledge Graph Embedding (KGE) model.

## 3 METHODOLOGY

In this section, we introduce the CNR-NST framework. We first describe the construction of the entity-numerical KGE model, followed by an explanation of how the pre-trained adjacency matrix is used for reasoning and computation in multi-hop queries. Figure 3 provides an overview of the CNR-NST framework.

## 3.1 Constructing Numerical and Entity Adjacency Matrices

**Pre-training Framework.** To address complex queries involving both entities and numerical values, we propose the Multi-ComplEx joint training framework (as shown in Figure 3), based on the ComplEx model (Trouillon et al. (2016)), and introduce three link predictors: $f(h, t, r) \in \mathbb{R}$, $f(h, n, a) \in \mathbb{A}$, and $f(v, w, np) \in \mathbb{F}$, which effectively capture relationships among entities, relations, attributes, and numerical values. The architecture designed to learn numerical size relationships is presented in Appendix A.2. These triples are jointly trained with the existing $(h, t, r)$ and $(h, n, a)$ triples from the knowledge graph training set, and the training objective function is as follows:

$$f(h, r, t) = \sum_{j \in \{\mathbb{R}, \mathbb{A}, \mathbb{F}\}} \beta_j Re\left(\langle h, r_j, \bar{t} \rangle\right) \tag{3}$$

where $\beta_j$ represents the the weight parameters under different types of triple relationships. For any given triple $(\mathbf{h}, \mathbf{t}, \mathbf{r})$, the loss function for link prediction training is defined as follows:

$$\mathcal{L}_X = - \sum_{(h,r,t) \in (\mathbb{R} \cup \mathbb{A} \cup \mathbb{F})} \log \sigma\left(f_X(h, r, t)\right) - \sum_{(h',r,t) \in (\mathbb{R}' \cup \mathbb{A}' \cup \mathbb{F}')} \log \sigma\left(f_X\left(h', r, t\right)\right) \tag{4}$$

$$\mathcal{L}_{joint} = \sum_{X \in (\mathbb{R} \cup \mathbb{A} \cup \mathbb{F})} \beta_X \cdot \mathcal{L}_X \tag{5}$$

Where $X$ represents various relationships between numerical attributes and entities, $\sigma$ denotes a normalization function, and $f_X$ is typically set as the cross-entropy loss function. The training details of the joint training framework are provided in Appendix A.2.

As demonstrated by the experiments in Section 4.4, the Multi-ComplEx joint training framework provides substantial improvements in numerical link prediction compared to methods that train entity and numerical relationships separately.

**Definition of the Scoring Function.** Using a Multi-ComplEx model, we can score the likelihood of three atomic formulas $f_r(e_i, e_j)$, $f_a(e_i, n_j)$, and $f_f(n_i, n_j)$. This is achieved by employing neural link predictors $f_E, f_A, f_N$ to infer missing edges in the knowledge graph in the $\mathcal{V} \times \mathcal{R} \times \mathcal{V}, \mathcal{V} \times \mathcal{A} \times \mathcal{N}, \mathcal{N} \times \mathcal{F} \times \mathcal{N}$ space. The probabilities of all known triples within the knowledge graph are set to 1, while the probability values of triples outside the known knowledge graph are modeled to follow the original distribution (Arakelyan et al. (2020)).We define the neural adjacency matrix $M_R \in [0,1]^{|\mathcal{R}| \times |\mathcal{V}| \times |\mathcal{V}|}$, $M_A \in [0,1]^{|\mathcal{A}| \times |\mathcal{V}| \times |\mathcal{N}|}$, $M_F \in [0,1]^{|\mathcal{F}| \times |\mathcal{N}| \times |\mathcal{N}|}$ as:

$$\hat{R}(x_i, x_j) = \frac{\exp\left(F(x_i, x_j) \cdot |\{(x_i, R, x) \in \mathcal{F}_{train} | x \in \mathcal{V} \cup \mathcal{N}\}|\right)}{\sum_{(()) x \in \mathcal{V} \cup \mathcal{N}} \exp\left(F(x_i, x)\right)} \tag{6}$$

$$\hat{R}(x_i, x_j) = \begin{cases} 1 & \text{if } (x_i, R, x_j) \in \mathcal{F}_{train} \\ \min\left\{\hat{R}(x_i, x_j), 1 - \epsilon\right\} & \text{otherwise} \end{cases} \tag{7}$$

where $x$ represent either a numerical value or an entity, and $R$ refers to one of the three relationships: $\mathcal{R}, \mathcal{A}, \mathcal{F}. \mathcal{F}_{train}$ represents the training set in the knowledge graph, and $x_i$ represents an entity or a numerical value. The expression $(x_i, R, x_j)$ refers to any triple in one of the three categories. The value of $\epsilon$ is typically set to 0.0001. Also note the matrix $M$ contains a large number of zeros, it will be stored as a sparse matrix, significantly reducing storage space.

## 3.2 Fuzzy Representation and Operator Definition

**Fuzzy set Representation.** CNR-NST utilizes fuzzy sets to address the inherent uncertainty of numerical values. Specifically, for an anchored entity $v$ or an anchored numerical value $n$, we represent them using an initialization vector $[0, 0, \ldots, 1, \ldots, 0]$, where the position corresponding to the entity or numerical value is set to 1, with all other positions set to 0. For intermediate entities or numerical values, we use fuzzy vectors $v_{1,\ldots,k} \in [0,1]^{|V|}$ ($n_{1,\ldots,j} \in [0,1]^{|N|}$), respectively, to represent their states. The uncertainty present in complex queries can then be quantified using a membership function.

$$\mathcal{U}(Q) = \{(x, \mu_A(x)) | x \in X\} \tag{8}$$

where element $x$ represents an entity or a numerical value, $\mathcal{U}(Q)$ denotes the probability that element $x$ satisfies the query $Q$, and the membership function is $\mu_A$.

**Numerical Operators.** There are three types of numerical operators: attribute projection, reverse attribute projection, and numerical projection. Let $V^*(X = x)$ represent the maximum truth value of a subquery rooted at $X$ when $x$ is assigned as an entity or numerical value. We recursively compute the $\mathbf{V}^*(X) = [V^*(X = x)]_{x \in \mathcal{V} \cup \mathcal{N}} \in [0,1]^{|N|}$ for each node in the query tree to maximize the overall truth value of the query tree, ultimately deriving the truth value $\mathbf{V}^*(x_?)$ at the root node.

When a node is connected to its child node by a edge, the fuzzy set of the node is calculated as follows:

$$\mathbf{V}^*(x_?) = \max_j \left( \left( V^*(x_k)^T \cdots \times |X| \right) \odot M_x \right) \tag{9}$$

where $max_j$ represents the maximum value in the column, and $x$ can represent either an entity or a numerical value. $M_x$ refers to the entity-value probability distribution matrix $M_a$, value-entity probability distribution matrix $M_a^T$ or the value-value probability distribution matrix $M_f$.

Numerical complex queries also involve entity-related operators, such as relation projection, intersection, and union. The mathematical formulations of these operators are provided in Appendix A.3 and will not be detailed here.

## 3.3 FUZZY REASONING AND COMPUTATION

In this section, we introduce how numerical complex queries are answered using the numerical atomic queries defined earlier. Starting from the anchor node $X_{anchor}$, which can represent either an entity or a numerical value, numerical reasoning and computations are executed based on the types of edges within the query computation tree.

**Numerical Reasoning.** At each step of numerical complex reasoning, CNR-NST assigns a score $\mathbf{V}^*(X_?) \in [0,1]^{|x|}$ to each node $X$, where this score reflects the likelihood of the query tree with $X_?$ as the endpoint being satisfied. CNR-NST then determines the optimal entity assignments by back-propagating through the query tree:

$$\mathbf{V}_i = \varphi_i(V_{i1}, V_{i2}, \cdots, V_{im}) \tag{10}$$

where $\mathbf{V}_i$ refers to the fuzzy representation of non-anchor nodes, and $\varphi$ denotes the various prediction functions applied to numerical values or entities. Specifically, when $V$ represents a numerical value, $V_{im} = \mu_m(n_m)$.

**Numerical Computation.** It is important to distinguish between numerical computation and numerical reasoning. Numerical computation involves performing arithmetic operations, such as addition, subtraction, multiplication, or division, on numerical values in the knowledge graph. In CNR-NST, when handling numerical complex queries, the numerical value at a given step is not a precise value but instead a fuzzy set $[0,1]^{|x|}$. We interpret the numerical values corresponding to the coordinates as membership functions, converting the fuzzy set into a fuzzy number.

Let $\mathcal{R}$ be the real number domain, and let the mapping $*: R \cdot R \to R$ be a binary operation on the real number domain (Mordeson (2001)). From this mapping, a new mapping $*: F(R) \cdot F(R) \to F(R)$ can be induced. Based on the extension principle in fuzzy mathematics, we can easily derive the following theorem:

$$\underline{A} \otimes \underline{B} = \int_R \bigvee_{x \otimes y = z} \left( \mu_{\underline{A}}(x) \bigwedge \mu_{\underline{B}}(y) \right) / z \tag{11}$$

Here, $\otimes$ can represent the four basic arithmetic operations on real numbers. For convenience, we often discretize the real number domain for processing, so the above expression is transformed into:

$$\underline{A} \otimes \underline{B} = \sum_z \frac{\bigvee_{x \otimes y = z} \left( \mu_{\underline{A}}(x) \bigwedge \mu_{\underline{B}}(y) \right)}{z} \tag{12}$$

As a result of the above derivation, we obtain a new fuzzy number and a membership set with a size of $|N|^2$. To prevent exponential growth in dimensionality during multiple numerical operations, we retain only the components with the highest membership values. This approach ensures that the dimensionality of the numerical fuzzy sets remains stable throughout the reasoning process.

Table 1: MRR (%) and MAE, MSE for the types of queries that LitCQD can support.

| Dataset | Method | 1rp | 2p | ppp | 2pi | 3p3i | 2pip | pppi | 2pu | 2pup |
|---|---|---|---|---|---|---|---|---|---|---|
| FB15K-237 | LitCQD | 34.76 | 11.43 | 7.38 | 37.47 | 43.71 | 16.66 | 23.52 | 4.25 | 4.55 |
| | CNR-NST | **35.36** | **18.11** | **15.94** | **43.53** | **59.02** | **27.55** | **36.11** | **16.55** | **16.63** |
| FB15K | LitCQD | 85.22 | 44.29 | 23.54 | 74.23 | 71.11 | 60.17 | 51.34 | 13.94 | 9.66 |
| | CNR-NST | **85.24** | **64.41** | **52.61** | **74.78** | **71.18** | **69.55** | **70.18** | **74.79** | **60.24** |
| DB15K | LitCQD | **38.55** | 27.71 | 19.01 | 59.17 | 74.04 | 36.85 | 53.84 | 11.34 | 16.78 |
| | CNR-NST | 36.89 | **30.01** | **24.18** | **60.47** | **76.46** | **38.45** | **54.25** | **24.29** | **27.23** |
| YAGO15K | LitCQD | 51.93 | 16.29 | 8.74 | 48.61 | **66.21** | 20.91 | 34.66 | 4.24 | 7.29 |
| | CNR-NST | **54.51** | **17.14** | **9.33** | **54.71** | 64.58 | **21.29** | **36.32** | **21.88** | **13.04** |
| Dataset | Method | nr | nrp | nrpi | 1ap(MAE) | 1ap(MSE) | pa(MAE) | pa(MSE) | 2pa(MAE) | 2pa(MSE) |
| FB15K-237 | LitCQD | 0.94 | 1.93 | 8.73 | 0.077 | 0.024 | 0.051 | 0.008 | 0.039 | 0.004 |
| | CNR-NST | **3.73** | **7.62** | **13.92** | **0.051** | **0.011** | **0.018** | **0.002** | **0.011** | **0.0007** |
| FB15K | LitCQD | 0.05 | 0.61 | 5.72 | 0.376 | 0.223 | 0.386 | 0.228 | 0.414 | 0.265 |
| | CNR-NST | **1.61** | **9.22** | **16.19** | **0.048** | **0.011** | **0.029** | **0.006** | **0.032** | **0.006** |
| DB15K | LitCQD | 0.19 | 1.63 | **31.68** | 0.042 | **0.015** | 0.039 | 0.008 | 0.039 | 0.006 |
| | CNR-NST | **1.48** | **3.64** | 14.59 | **0.042** | 0.017 | **0.017** | **0.003** | **0.013** | **0.001** |
| YAGO15K | LitCQD | 0.14 | 0.99 | **16.35** | **0.049** | **0.007** | 0.062 | 0.011 | 0.079 | 0.013 |
| | CNR-NST | **2.67** | **1.45** | 8.55 | 0.061 | 0.013 | **0.061** | **0.010** | **0.058** | **0.008** |

## 3.4 DISCUSSION

**Space Complexity.** The memory usage of CNR-NST includes the composite neural adjacency matrix $M_x$, which contains $|V^2||\mathcal{R}| + |N^2||\mathcal{F}| + |V||N||\mathcal{A}|$ elements. However, as described in Section 4.1, most values in the neural matrix $M_x$ can be filtered by a threshold $\epsilon$. Experimental results show that $M_x$ can be stored on a single GPU.

**Time Complexity.** In the numerical reasoning process, each variable is computed with a complexity of $O\left(|\mathcal{X}^2|\right)$, (where $\mathcal{X}$ refers to either numerical values or entities) but since each variable contains a large number of zero elements, the actual complexity of a single query, as derived from Equation 6, is $O\left(|\mathcal{X}| \cdot N \cdot |\mathbf{V}^*\left(X_k\right) > 0|\right)$, where $N$ represents the number of projections involved in the query.

## 4 EXPERIMENT

### 4.1 EXPERIMENTAL SETUP

**Dataset.** We conducted experiments on three datasets: FB15K, DB15K, and YAGO15K(Kotnis & García-Durán (2018)). Details regarding these datasets can be found in Appendix B.3. We used the 8 major categories and 92 subtypes of multi-hop numerical queries introduced by NRN (Bai et al. (2023a)). A full statistical overview of these numerical queries is available in Appendix B.4.

**Baselines.** We selected the LitCQD (Demir et al. (2023)) model and the three numerical reasoning models from NRN (Bai et al. (2023a)) as our baselines. However, because the calculation of average MRR values in NRN relies heavily on the query sampling method, and their original dataset did not include all 92 query types, we expanded the dataset accordingly. We took into account the differences in average calculation methods and the number of queries, and conducted comprehensive comparative experiments to ensure consistency and accuracy.

**Evaluation Protocol.** For each numerical complex query, the answers are classified as either easy or hard, depending on whether they can be directly inferred from existing edges in the graph. For example, in the test set, easy answers are those that can be derived from the training and validation graphs, whereas hard answers require reasoning over missing edges from the training and validation graphs. We adhere to the standard evaluation metrics used for complex multi-hop queries, which include Mean Reciprocal Rank (MRR) and Hits@K on the test set (Zhang et al. (2019)).

**Implementation Details.** We initially trained several KGE models on the training graph, using an extended version of the Multi-ComplEx model, which is based on ComplEx (Trouillon et al. (2016))

Table 2: MRR Results (%) for All Query Types in the Test Set; Query_N represents the total number of query types, while AVG_M refers to the method used to calculate the average. Avg_W indicates the weighted average based on the number of query types, and Avg represents the direct average of each subclass. For detailed results, see Appendix D.1.

| Dataset | Query_N | Avg_M | Method | Avg_All | 1p | 2p | 2i | 3i | pi | ip | 2u | up | 2b | 3b | bp | pb | 2pb |
|---|---|---|---|---|---|---|---|---|---|---|---|---|---|---|---|---|---|
| FB15K | | Avg_W | GQE+NRN | 20.04 | 31.69 | 8.15 | 33.99 | 41.19 | 23.10 | 10.10 | 7.57 | 4.51 | - | - | - | - | - |
| | | | Q2B+NRN | 22.36 | 37.01 | 7.74 | 36.88 | 42.72 | 24.25 | 10.09 | 15.61 | 4.54 | - | - | - | - | - |
| | | | Q2P+NRN | 24.44 | 42.75 | 12.87 | 33.71 | 38.75 | 23.14 | 13.23 | 23.16 | 7.89 | - | - | - | - | - |
| | 77 | | CNR-NST | **40.78** | **60.26** | **38.35** | **51.49** | **54.01** | **36.89** | **36.16** | **24.91** | **24.19** | - | - | - | - | - |
| | | Avg | GQE+NRN | 11.76 | 11.19 | 4.24 | 18.85 | 34.66 | 7.52 | 11.47 | 3.41 | 2.73 | - | - | - | - | - |
| | | | Q2B+NRN | 12.83 | 14.33 | 3.87 | 21.08 | 34.36 | 8.13 | 12.74 | 5.02 | 3.09 | - | - | - | - | - |
| | | | Q2P+NRN | 11.62 | 12.20 | 3.31 | 19.43 | 33.57 | 5.52 | 10.85 | 5.52 | 2.56 | - | - | - | - | - |
| | | | CNR-NST | **22.93** | **25.82** | **15.51** | **29.84** | **43.07** | **21.52** | **23.80** | **11.67** | **12.20** | - | - | - | - | - |
| | | Avg | GQE+NRN | 15.05 | 9.79 | 4.59 | 27.83 | 45.44 | 8.67 | 17.91 | 3.32 | 2.85 | - | - | - | - | - |
| | 92 | | Q2B+NRN | 16.20 | 12.04 | 4.16 | 28.21 | **48.07** | 18.93 | 9.99 | 5.02 | 3.14 | - | - | - | - | - |
| | | | Q2P+NRN | 10.34 | 9.65 | 3.47 | 17.50 | 26.41 | 11.96 | 5.47 | 5.65 | 2.58 | - | - | - | - | - |
| | | | CNR-NST | **24.28** | **27.14** | **19.79** | **30.44** | 31.07 | **28.42** | **24.17** | **17.38** | **15.85** | - | - | - | - | - |
| | 102 | | CNR-NST | **19.88** | **27.14** | **19.79** | **30.44** | 31.07 | 28.42 | 24.17 | 17.38 | 15.85 | **11.16** | **11.51** | **10.29** | **18.96** | **12.24** |
| DB15K | | Avg_W | GQE+NRN | 10.96 | 10.29 | 2.53 | 20.14 | 35.46 | 12.50 | 2.52 | 2.08 | 2.14 | - | - | - | - | - |
| | | | Q2B+NRN | 11.89 | 10.96 | 2.71 | 22.60 | 37.44 | 13.81 | 3.05 | 2.41 | 2.13 | - | - | - | - | - |
| | | | Q2P+NRN | 12.98 | 14.71 | 3.81 | 23.75 | 36.66 | 14.47 | 2.96 | 4.63 | 2.81 | - | - | - | - | - |
| | 77 | | CNR-NST | **23.46** | **22.54** | **16.98** | **36.60** | **46.59** | **30.67** | **14.91** | **8.40** | **10.96** | - | - | - | - | - |
| | | Avg | GQE+NRN | 10.91 | 3.50 | 2.72 | 17.60 | 43.30 | 11.28 | 5.54 | 1.40 | 1.91 | - | - | - | - | - |
| | | | Q2B+NRN | 12.01 | 4.15 | 2.78 | 19.12 | 47.88 | 12.53 | 6.31 | 1.36 | 1.98 | - | - | - | - | - |
| | | | Q2P+NRN | 11.90 | 4.72 | 3.01 | 13.52 | **50.03** | 13.81 | 5.21 | 2.51 | 2.41 | - | - | - | - | - |
| | | | CNR-NST | **15.85** | **10.93** | **8.42** | **23.38** | 45.61 | **17.17** | **11.09** | **4.46** | **5.77** | - | - | - | - | - |
| | | Avg | GQE+NRN | 14.73 | 3.33 | 2.99 | 23.47 | **58.45** | 19.14 | 7.12 | 1.38 | 1.97 | - | - | - | - | - |
| | 92 | | Q2B+NRN | 15.26 | 3.92 | 3.21 | 25.16 | 57.24 | 20.83 | 8.13 | 1.41 | 2.19 | - | - | - | - | - |
| | | | Q2P+NRN | 10.90 | 4.71 | 2.99 | 18.54 | 37.47 | 13.98 | 4.84 | 2.46 | 2.20 | - | - | - | - | - |
| | | | CNR-NST | **18.13** | **11.12** | **8.56** | **25.31** | 47.58 | **24.61** | **12.00** | **7.95** | **7.94** | - | - | - | - | - |
| | 102 | | CNR-NST | **16.38** | **11.12** | **8.56** | **25.31** | 47.58 | 24.61 | 12.00 | 7.95 | 7.94 | **12.27** | **11.39** | **5.72** | **21.49** | **16.94** |
| YAGO15K | | Avg_W | GQE+NRN | 15.60 | 14.79 | 4.23 | 35.68 | 39.94 | 18.29 | 5.65 | 4.23 | 1.96 | - | - | - | - | - |
| | | | Q2B+NRN | 18.78 | 21.40 | 4.59 | 39.72 | 45.16 | 19.62 | 7.90 | 9.05 | 2.82 | - | - | - | - | - |
| | | | Q2P+NRN | 14.82 | 22.97 | 5.70 | 25.70 | 29.04 | 14.38 | 5.40 | **11.87** | 3.51 | - | - | - | - | - |
| | 77 | | CNR-NST | **27.71** | **31.11** | **16.32** | **44.30** | **54.98** | **33.64** | **21.48** | 10.13 | **9.71** | - | - | - | - | - |
| | | Avg | GQE+NRN | 15.53 | 2.72 | 3.20 | 26.72 | 62.76 | 18.01 | 6.68 | 1.70 | 2.47 | - | - | - | - | - |
| | | | Q2B+NRN | 16.00 | 4.14 | 3.10 | 26.30 | 62.50 | 20.83 | 6.81 | 1.88 | 2.45 | - | - | - | - | - |
| | | | Q2P+NRN | 17.99 | 7.80 | 3.73 | 36.62 | 59.57 | 22.35 | 7.54 | 3.35 | 2.96 | - | - | - | - | - |
| | | | CNR-NST | **26.00** | **19.55** | **11.05** | **45.58** | **68.13** | **34.74** | **16.45** | **6.31** | **6.20** | - | - | - | - | - |
| | | Avg | GQE+NRN | 16.92 | 2.59 | 3.48 | 31.13 | 62.28 | 25.19 | 6.63 | 1.61 | 2.47 | - | - | - | - | - |
| | 92 | | Q2B+NRN | 17.64 | 3.20 | 3.03 | 32.52 | **65.53** | 25.53 | 7.40 | 1.60 | 2.34 | - | - | - | - | - |
| | | | Q2P+NRN | 16.50 | 6.21 | 3.71 | 28.30 | 53.39 | 25.51 | 8.58 | 3.28 | 3.03 | - | - | - | - | - |
| | | | CNR-NST | **22.97** | **15.50** | **11.24** | **38.74** | 55.22 | **34.23** | **16.44** | **5.90** | **6.45** | - | - | - | - | - |
| | 102 | | CNR-NST | **19.51** | **15.50** | **11.24** | **38.74** | 55.22 | 34.23 | 16.44 | 5.90 | 6.45 | **11.71** | **12.49** | **5.92** | **18.94** | **20.79** |

Table 3: MRR and MRR_0.001 Results (%) for the Numerical Queries. For detailed results, see Appendix D.2.

| Dataset | Eva_metric | Method | Avg_All | 1p | 2p | 2i | 3i | pi | ip | 2u | up |
|---|---|---|---|---|---|---|---|---|---|---|---|
| FB15K | MRR | Q2P+NRN | 4.91 | 0.42 | 0.86 | 9.06 | 17.75 | 7.39 | 3.17 | 0.14 | 0.47 |
| | | CNR-NST | **20.01** | **17.77** | **15.33** | **27.87** | **37.84** | **28.70** | **18.38** | **4.25** | **9.91** |
| | MRR_0.001 | Q2P+NRN | 1.11 | 0.45 | 0.57 | 1.44 | 3.52 | 1.22 | 0.46 | 0.34 | 0.90 |
| | | CNR-NST | **24.94** | **22.54** | **16.41** | **36.26** | **46.98** | **37.83** | **22.61** | **7.66** | **9.26** |
| DB15K | MRR | Q2P+NRN | 2.91 | 0.22 | 0.49 | 6.56 | 10.25 | 3.29 | 2.12 | 0.06 | 0.31 |
| | | CNR-NST | **12.84** | **6.77** | **5.93** | **17.28** | **26.64** | **21.03** | **10.17** | **7.81** | **7.07** |
| | MRR_0.001 | Q2P+NRN | 10.67 | 4.96 | 6.29 | 4.58 | 5.60 | 9.66 | 13.69 | 18.28 | 22.27 |
| | | CNR-NST | **24.79** | **18.92** | **15.45** | **25.62** | **31.61** | **35.88** | **24.90** | **23.24** | **22.70** |
| YAGO15K | MRR | Q2P+NRN | 10.63 | 0.51 | 1.17 | 20.31 | 36.20 | 16.97 | 8.24 | 0.31 | 1.31 |
| | | CNR-NST | **17.48** | **12.31** | **11.80** | **25.16** | **37.16** | **25.87** | **19.38** | **3.16** | **5.02** |
| | MRR_0.001 | Q2P+NRN | 19.26 | 5.66 | **21.09** | 9.48 | 10.42 | 13.06 | 21.80 | **39.05** | **33.52** |
| | | CNR-NST | **25.36** | **18.35** | 17.39 | **32.92** | **37.59** | **34.1** | **22.37** | 23.24 | 16.91 |

and employs N3 regularization (Lacroix et al. (2018)). Next, we derived several computational neural adjacency matrices $M_x$ from these KGE models. To minimize memory consumption for $M_x$, we applied an adjacency matrix filter $A$ to eliminate values below a certain threshold, converting the matrix into a sparse format suitable for storage on a single NVIDIA A40 GPU.

## 4.2 Results on Numerical Complex Query Answering

Tables 1 and 2 demonstrate the performance of CNR-NST in complex numerical query reasoning across three public datasets. In Table 1, we provide a comprehensive comparison of CNR-NST with the LitCQD model on query types supported by LitCQD across four public datasets. The results indicate that CNR-NST supports a wider range of query types and generally outperforms LitCQD in performance. Additionally, Table 2 presents a thorough comparison between CNR-NST and three benchmark models. Notably, even without specialized training for various complex query types, CNR-NST consistently outperforms the baseline models across almost all query types, with performance improvements exceeding 100% in many cases. Furthermore, as shown in Table 3, CNR-NST achieves remarkable improvements in queries with numerical answers, with an average performance increase of over 300We attribute this improvement to the use of a pre-training framework that predicts the link probabilities between entities and numerical values, thereby enhancing the generalization capability for complex multi-hop queries (Section 4.4 has experimental verification). The multiple relationships between entities and numerical values act as additional constraints during numerical reasoning, reducing the reasoning space and improving accuracy. Moreover, fuzzy sets naturally manage the numerical fuzziness inherent in complex reasoning, further enhancing CNR-NST's accuracy in these tasks.

## 4.3 Answering Complex Numerical Computation Query

The 92 query types mentioned earlier are still restricted to reasoning with individual numerical values. In this paper, we introduce five new query types that involve calculations and reasoning with two or more numerical values within the knowledge graph, further increasing the diversity of multi-hop queries. For the first time, we extend numerical reasoning in knowledge graphs to the real number domain, whereas previous methods were confined to the discrete numerical domain within the KG. This extension enables more precise modeling of real-world knowledge graph queries.

As shown in Table 2, we define five new query types: 2b, 3b, bp, pb, and 2pb. The $B$ operator represents binary operations (addition, subtraction, multiplication, and division) performed on two numerical values. These query types result in numerical subqueries, and because their answers are not present in the original knowledge graph, we use the new evaluation metrics introduced in Section 4.4. Detailed descriptions of these query types are provided in Appendix E.

## 4.4 Supplementary Experimental Details

**New Evaluation Metrics for Numerical-Type Answers.** In complex multi-hop numerical reasoning, many query types return numerical values as answers. Previous approaches used the same evaluation metrics for these queries as for entities, but this method has limitations. It does not consider the differences between numerical values and entities, especially the continuous nature of numerical data. Additionally, it cannot effectively evaluate queries where the numerical answers are not present in the original knowledge graph. To address these issues, we propose a new evaluation metric for numerical answers, analogous to Mean Reciprocal Rank (MRR). Instead of ranking based on the exact match of numerical nodes, we compute the RANK using the probability ranking of numerical nodes whose relative error compared to the correct answer is below a specified threshold (typically set at 0.001). This new metric is denoted as $MRR_{0.001}$, and the ranking calculations are applied only to hard answers.

As shown in Table 3, we re-evaluated the numerical answers across the 102 sub-queries that CNR-NST can handle and compared the results with the baseline model, Q2P. CNR-NST significantly outperformed Q2P on the new evaluation metric, demonstrating that although CNR-NST may not always generate completely accurate answers in numerical reasoning, its performance improves substantially when a margin of error is permitted. This suggests that CNR-NST's inferred answers are often very close to the correct values. Furthermore, the experimental results show that our approach offers a significant advantage over baseline models in numerical reasoning tasks, with an average performance improvement of 200%.

**Ablation Study of the Pre-training Framework.** As shown in Table 4, we conducted an ablation study within the pre-training framework. We experimented with the single ComplEx model, two configurations of the ComplEx model, and the Multi-ComplEx model without accounting for

Table 4: MRR Results (%) for the Predictors (ERE, EAV, VFV) on the Validation and Test Sets

| Dataset | Pre-training Framework | Avg_Test | ERE_Test | EAV_Test | VFV_Test |
|---------|-----------------------|----------|----------|----------|----------|
| FB15K | ComplEx | 18.54 | 25.96 | 5.51 | 24.15 |
| | Two ComplEx | 17.86 | 27.17 | 1.92 | 24.48 |
| | Multi-ComplEx W/O EC | 34.50 | **27.58** | 3.02 | **72.90** |
| | Multi-ComplEx | **35.36** | 26.73 | **6.78** | 72.58 |
| DB15K | ComplEx | 21.42 | 29.22 | 6.97 | 28.08 |
| | Two ComplEx | 21.25 | 30.72 | 5.72 | 27.32 |
| | Multi-ComplEx W/O EC | 33.00 | **30.90** | 5.16 | 62.95 |
| | Multi-ComplEx | **46.06** | 30.55 | **11.26** | **96.36** |
| YAGO15K | ComplEx | 11.59 | 11.46 | 1.98 | 21.32 |
| | Two ComplEx | 11.14 | 10.68 | 1.19 | 21.55 |
| | Multi-ComplEx W/O EC | 30.75 | 10.88 | 1.05 | 80.33 |
| | Multi-ComplEx | **32.13** | **11.48** | **3.39** | **81.53** |

Table 5: Training and Inference Times on the FB15K Dataset (Training Time in Hours, Inference Time in ms/query)

| Method | Training_time | 1p | 2p | 2i | 3i | pi | ip | 2u | up | 2b | 3b | bp | pb | 2pb |
|--------|---------------|------|-------|------|-------|-------|-------|------|-------|------|------|--------|-------|--------|
| Q2P+NRN | 32.40 | 0.13 | 0.14 | 0.25 | 0.33 | 0.24 | 0.26 | 0.28 | 0.29 | - | - | - | - | - |
| CNR-NST | 0.22 | 3.06 | 15.83 | 4.53 | 10.09 | 21.19 | 16.66 | 4.54 | 39.51 | 6.07 | 7.75 | 603.78 | 80.81 | 528.20 |

the numerical associations of entities (Multi-ComplEx W/O EC). The results demonstrate that the Multi-ComplEx model significantly outperforms the ComplEx model in numerical prediction tasks. Additionally, the knowledge embedded in the relationships between entities greatly enhances the accuracy of numerical predictions and improves the performance of complex numerical queries.

**Training and Inference Efficiency.** As shown in Table 5, during the training phase, CNR-NST only requires training on single-hop queries involving numerical values and entities, resulting in shorter training times compared to NRN, which must train on the entire dataset. CNR-NST's inference process requires calling a pre-trained prediction model at each step of the reasoning chain. In contrast, NRN has fixed neural network dimensions, which allows for faster inference. However, even on a dataset with 400,000 queries, CNR-NST completes the inference in just four hours, which is still significantly less time than what NRN requires for training.

## 5 RELATED WORK

**Complex Query Answering over conventional Knowledge Graph.** Complex Query Answering (CQA) focuses on reasoning over relationships and entities in a knowledge graph to address complex logical queries. Early CQA methods relied on logical reasoning rules to query knowledge graphs. Methods like GQE (Hamilton et al. (2018)), Q2B (Ren et al. (2020)), Q2P (Bai et al. (2022)), ConE (Zhang et al. (2021)) and BetaE (Ren & Leskovec (2020)) represent entity sets and queries using geometric shapes or probability distributions, utilizing geometric operations (e.g., intersections, projections) or probabilistic operations for reasoning. Later, CQD (Arakelyan et al. (2020)) introduced a framework capable of handling complex logical expressions without explicit training on complex queries. QTO (Bai et al. (2023b)) further optimized the query computation tree during complex query processing, improving reasoning accuracy and reducing the search space.

**Numerical reasoning over Knowledge Graph.** Numerical reasoning tasks in knowledge graphs involve making logical inferences or predictions based on numerical values associated with entities and relationships. Methods such as RAKGE (Kim et al. (2023)) , KR-EAR (Lin et al. (2016)) , TransEA Wu & Wang (2018) and Lacroix et al. (2018) utilize attribute learning to improve numerical reasoning within knowledge graph embeddings. LiteralE (Kristiadi et al. (2019)) enhances knowledge graph embeddings by incorporating textual information through learnable parametric functions. HyNT (Chung et al. (2023)) uses the expressive power of Transformers to capture complex relational structures and numerical attributes in knowledge graphs. Neural-Num-LP (Wang et al. (2020)) learns numerical rules within knowledge graphs.

**LLMs' numerical reasoning.** Large model numerical reasoning refers to extracting relevant numerical information from textual descriptions and performing mathematical calculations (Zhang et al. (2024)). MathPrompter (Imani et al. (2023)) introduces the "Chain-of-Thought" (CoT) approach,

utilizing a step-by-step prompting technique to guide the model through solving complex arithmetic problems incrementally. NumeroLogic (Schwartz et al. (2024)) defines new numerical formats to handle and execute arithmetic operations. PoT (Chen et al. (2022)) employs the Codex model to represent the reasoning process as a program, which is then executed by an external system to perform the necessary computations and derive the final answer.

**Complex Query Answering over Numerical Knowledge Graph.** Research on CQA with numerical values is still limited. LitCQD (Demir et al. (2023)) decomposes complex numerical queries into smaller subqueries, solving them by integrating symbolic and numerical reasoning. NRN (Bai et al. (2023a)) embeds entities, relations, and numerical attributes from the knowledge graph into a shared vector space, using neural networks to learn and reason about numerical relationships.

# 6 CONCLUSION

In this paper, we present a novel framework for reasoning and computation on numerical knowledge graphs, leveraging a pretrained multi-relational link predictor to infer over 102 types of complex numerical queries. Extensive experiments on three publicly available KGs demonstrate that our proposed model, CNR-NST, significantly surpasses previous SOTA methods. Furthermore, we introduce additional categories of numerical reasoning tasks and new evaluation metrics for numerical answers, contributing to the broader research of multi-hop numerical reasoning. Future work could explore expanding the variety of numerical queries supported by CNR-NST, further enhancing its applicability in diverse reasoning tasks.

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

CONTENTS

# A   MATHEMATICAL PROOF

## A.1   FUZZY SET-BASED NUMERICAL COMPUTATIONS

This section will demonstrate how mappings in the set of real numbers can be induced into mappings between fuzzy sets.

First, we explain the concept of induction. Let there be a mapping on the real number domain:

$$f : U \to V$$

$$u \mapsto v = f(u)$$

From this, a new mapping can be induced, which we still denote as $f$:

$$f : P(U) \to P(V)$$

$$A \mapsto B = f(A)$$

$$f(A) \triangleq \{v | \exists u \in A, let\ f(u) = v, v \in V\}$$

The key to how a mapping on a set of real numbers can induce a mapping between fuzzy sets lies in determining the membership function of the fuzzy set.The definition of the extension principle is given below: Let $f : U \to V$, From $f$, we can induce two mappings:

$$f : F(U) \to F(V), f^{-1} : F(V) \to F(U)$$

$$\underset{\sim}{A} \mapsto f\left(\underset{\sim}{A}\right), \underset{\sim}{B} \mapsto f^{-1}\left(\underset{\sim}{B}\right)$$

The membership functions of the induced mappings are as follows:

$$\mu_{f(A_\sim)}(v) = \begin{cases} \bigvee_{f(u)=v} \mu_{A_\sim}(u) & \text{if } \exists u \in U, let\ f(u) = v \\ 0 & \text{otherwise} \end{cases}$$

$$\mu_{f^{-1}(B)}(u) = \mu_{B_\sim}(v), v = f(u)$$

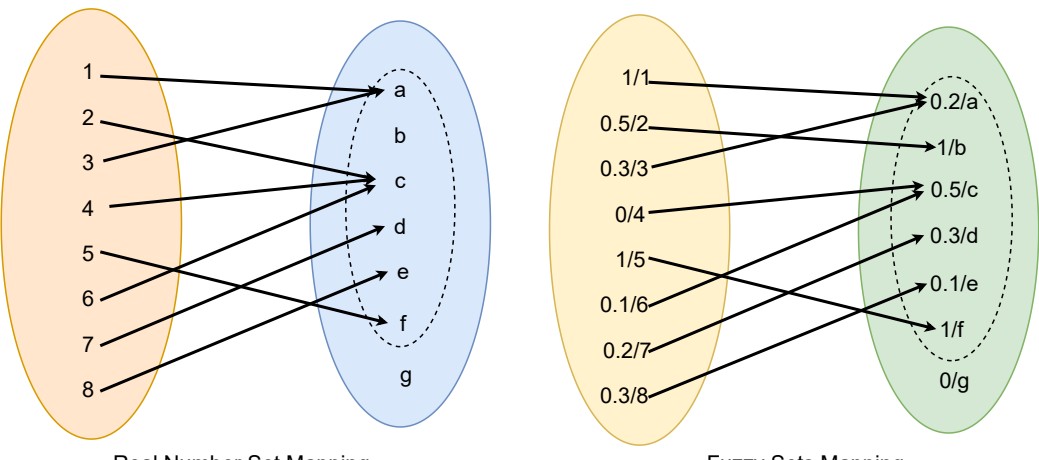

Figure 4: Visualization of the Extension Principle

## A.2 DETAILS OF THE TRAINING PROCEDURE FOR THE MULTI-COMPLEX

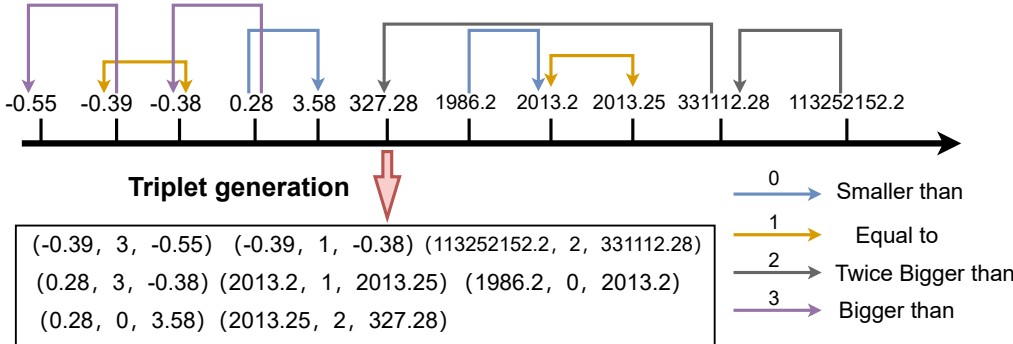

Figure 5: Numerical relationship triplet generation based on the values in the training set for pre-training of the model.

During the joint pre-training stage of CNR-NST, the objective function of the Multi-ComplEx model is defined as follows:

$$f_A\left(h, r, t\right) = Re\left(\langle \mathbf{h}, \mathbf{r_A}, \bar{\mathbf{t}} \rangle\right)$$

$$f_B\left(h, r, v\right) = Re\left(\langle \mathbf{h}, \mathbf{r_B}, \bar{\mathbf{v}} \rangle\right)$$

$$f_C\left(m, r, n\right) = Re\left(\langle \mathbf{m}, \mathbf{r_C}, \bar{\mathbf{n}} \rangle\right)$$

where $h, t$ denotes the entity embeddings, $r$ refers to the embeddings for different types of relations, and $v, m, n$ represents the numerical embeddings.

We employ cross-entropy loss as the optimization objective. For each triplet $(h, r, t)$, the loss function for each predictor is defined as follows:

$$\mathcal{L}_A = - \sum_{(h,r,t)\in\mathbb{R}} \log\sigma\left(f_A\left(h, r, t\right)\right) + \sum_{\left(h^{'},r,t\right)\in\mathbb{R}^{'}} \log\sigma\left(-f_A\left(h^{'}, r, t\right)\right)$$

$$\mathcal{L}_B = - \sum_{(h,r,n)\in\mathbb{A}} \log\sigma\left(f_B\left(h, r, n\right)\right) + \sum_{\left(h^{'},r,n\right)\in\mathbb{A}^{'}} \log\sigma\left(-f_B\left(h^{'}, r, n\right)\right)$$

$$\mathcal{L}_C = - \sum_{(m,r,n)\in\mathbb{F}} \log\sigma\left(f_C\left(m, r, n\right)\right) + \sum_{\left(m^{'},r,n\right)\in\mathbb{F}^{'}} \log\sigma\left(-f_C\left(m^{'}, r, n\right)\right)$$

The loss functions of the three predictors are weighted and combined to form a joint loss function, where the weights are denoted by $W_{EAV}$ and $W_{VFV}$. These weights can be adjusted based on the task's requirements and importance. The joint loss function is defined as follows:

$$\mathcal{L}_{joint} = \mathcal{L}_A + W_{EAV} \cdot \mathcal{L}_B + W_{VFV} \cdot \mathcal{L}_C$$

To prevent overfitting, regularization terms are often incorporated into the loss function. L1 and L2 regularization can be applied to control the magnitude of the model parameters. For each predictor, the regularization term is defined as follows:

$$\Omega = \lambda\left(\|\mathbf{h}\|^2 + \|\mathbf{r_A}\|^2 + \|\mathbf{t}\|^2 + \|\mathbf{r_B}\|^2 + \|\mathbf{v}\|^2 + \|\mathbf{m}\|^2 + \|\mathbf{r_C}\|^2 + \|\mathbf{n}\|^2\right)$$

The final joint objective function incorporates both the loss function and the regularization terms, and is given by the following expression:

$$\mathcal{L} = \mathcal{L}_{joint} + \Omega$$

A.3 MATHEMATICAL REPRESENTATION OF MULTI-HOP LOGICAL REASONING

This section will use mathematical expressions to explain and infer various logical operators and mappings involved in complex numerical reasoning.

Let $V^*\left(X_? = x\right)$ represent the maximum truth value of the subquery at the root node $x$ when node $X$ is assigned a value $x$.

Assuming that the root node $x$ is formed by merging sub-node $\left\{x_?^1, \ldots, x_?^K\right\}$ by **intersection**, the maximum truth value of the query is given by the following expression:

$$V^*\left(x_? = e\right) = \top_{1 \le i \le K} V^*\left(x_? = e\right)$$

$$V^*\left(x_? = n\right) = \top_{1 \le i \le K} V^*\left(x_? = n\right)$$

$$\Rightarrow \mathbf{V}^*\left(x_?\right) = \prod_{1 \le i \le K} \mathbf{V}^*\left(x_?^i\right)$$

Similarly, when the root node is merged by **union**, the following expression applies:

$$V^*\left(x_? = e\right) = \bot_{1 \le i \le K} V^*\left(x_? = e\right)$$

$$V^*\left(x_? = n\right) = \bot_{1 \le i \le K} V^*\left(x_? = n\right)$$

$$\Rightarrow \mathbf{V}^*\left(x_?\right) = 1 - \prod_{1 \le i \le K} \left(1 - \mathbf{V}^*\left(x_?^i\right)\right)$$

When the root node is connected to its child edges through any relational edge, the maximum truth value of the query with this node as the root is expressed as shown in Equation 9 in the main text. Depending on the type of relation, the expression has the following four variations:

$$\mathbf{V}^*\left(v_?\right) = \max_j \left(\left(V^*\left(v_k\right)^T \cdots \times |v|\right) \odot M_E\right)$$

$$\mathbf{V}^*\left(n_?\right) = \max_j \left(\left(V^*\left(v_k\right)^T \cdots \times |v|\right) \odot M_A\right)$$

$$\mathbf{V}^*\left(v_?\right) = \max_j \left(\left(V^*\left(n_k\right)^T \cdots \times |n|\right) \odot M_A^{-1}\right)$$

$$\mathbf{V}^*\left(n_?\right) = \max_j \left(\left(V^*\left(n_k\right)^T \cdots \times |n|\right) \odot M_F\right)$$

# B DATASET STATISTICS

## B.1 NUMERICAL SPARSITY OF THE DATASET.

In this section, we present the numerical sparsity in the three public datasets FB15K, DB15K, and YAGO15K.

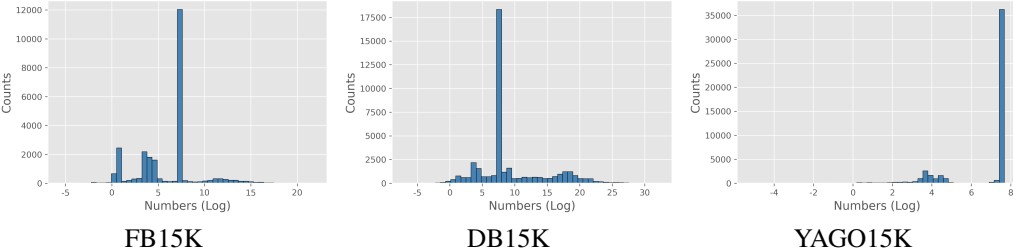

FB15K       DB15K       YAGO15K

Figure 6: Numerical Distribution Across the Three Public Datasets

As shown in the three figures above, the horizontal axis represents the logarithmic transformation of actual values, while the vertical axis shows the frequency of occurrences. We observe that most values are concentrated in a specific region, yet the overall range of values remains broad, highlighting the numerical sparsity characteristic of numerical knowledge graphs.

## B.2 KNOWLEDGE GRAPH CONTRUCTION

Table 5 presents the detailed information for constructing knowledge graphs from the original datasets, from which we will randomly sample to form queries.

Table 6: The statistics of the three knowledge graphs

| Dataset | Data Split | Nodes_N | Rel_N | Attr_N | Rel_Edges | Attr_Edges | Num_Edges |
|---|---|---|---|---|---|---|---|
| FB15K | Training | 25106 | | | 947540 | 20248 | 27020 |
| | Validation | 26108 | 1345 | 15 | 1065982 | 22779 | 27376 |
| | Testing | 27144 | | | 1184426 | 25311 | 27389 |
| DB15K | Training | 31980 | | | 145262 | 33131 | 25495 |
| | Validation | 34191 | 279 | 30 | 161978 | 37269 | 25596 |
| | Testing | 36358 | | | 178394 | 41411 | 25680 |
| YAGO15K | Training | 32112 | | | 196616 | 21732 | 26616 |
| | Validation | 33078 | 32 | 7 | 221194 | 22748 | 26627 |
| | Testing | 36358 | | | 245772 | 23520 | 26631 |

## B.3 DATASET INFORMATION FOR PRE-TRAINING

Table 6 shows the training dataset information used for our pre-training framework Multi-ComplEx. The VFV triples are as defined by us, and the dataset splits are identical to those in Table 5.

Table 7: Details of the Datasets Used for Pre-training

| Dataset | Data Split | ERE Triad | EAV Triad | VFV Triad |
|---|---|---|---|---|
| FB15K | Training | 473770 | 20248 | 220421 |
| | Validation | 59221 | 2531 | 27553 |
| | Testing | 59222 | 2532 | 27553 |
| DB15K | Training | 79222 | 33145 | 296006 |
| | Validation | 9903 | 4143 | 37001 |
| | Testing | 9903 | 4144 | 37001 |
| YAGO15K | Training | 98308 | 18816 | 229804 |
| | Validation | 12289 | 2352 | 28725 |
| | Testing | 12289 | 2352 | 28726 |

## B.4 SAMPLE QUERIES FROM GRAPH

Figure 7 provides a visual representation of the structure for most of the 102 query types. Tables 7 and 8 present the distribution of 13 major query types and 102 subquery types sampled from the three datasets. Note that these quantities only include the test set. Since CNR-NST does not require training on complex queries, we only sample from the test set and conduct experiments. Supplementary explanation: in the Query_Name column of the table, "p" stands for relation prediction, "ap" stands for attribute prediction, abbreviated as "a", "np" stands for numerical prediction, abbreviated as "n", and "rap" stands for reverse attribute prediction, abbreviated as "r". Table 9 shows the percentage of inference paths present on the top edge of the test set for each query type sampled on our dataset, reflecting to some extent the difficulty of the queries we sampled.

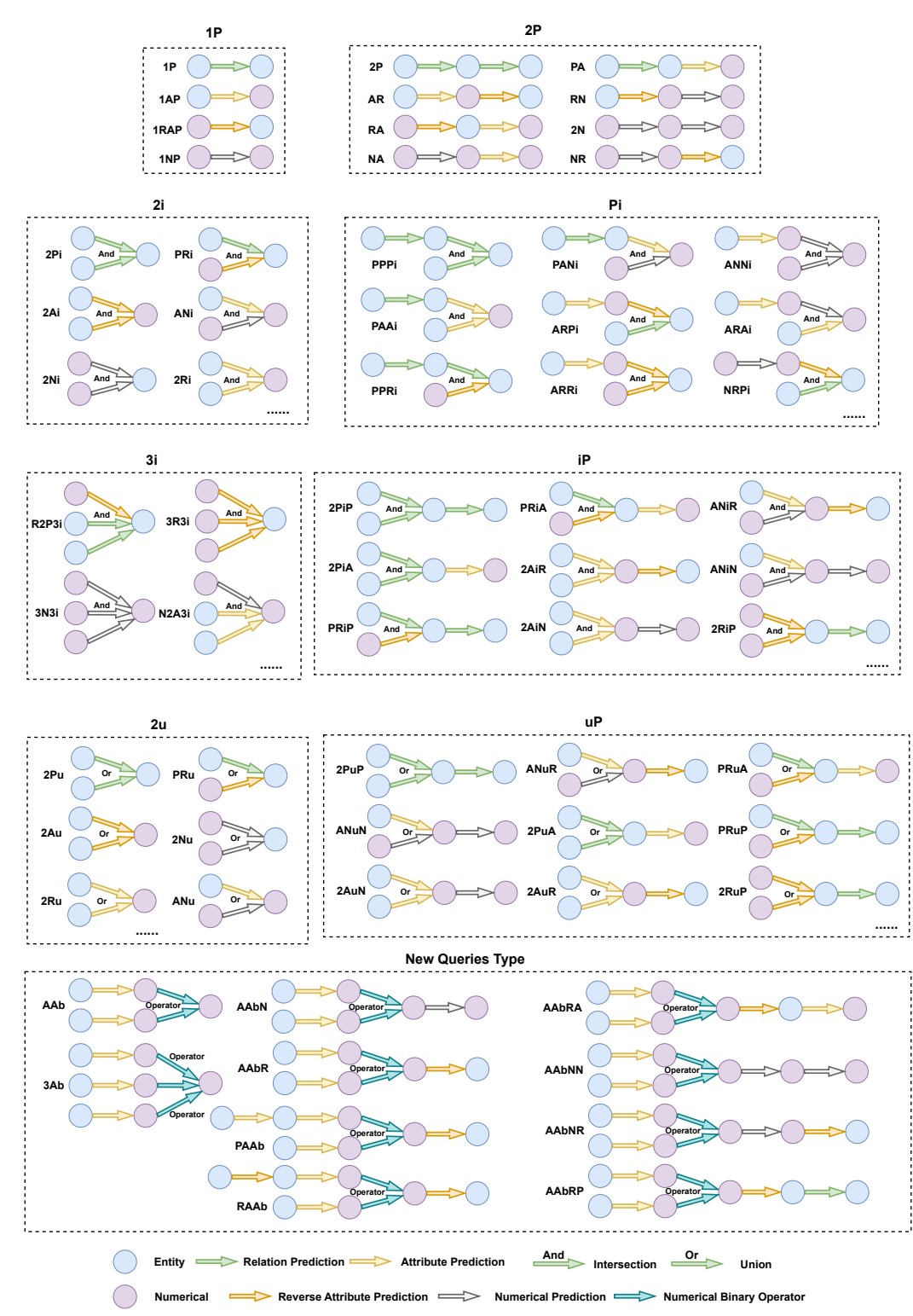

Figure 7: The visualization of complex numerical reasoning structures includes a total of 102 query types, of which we have selected only a subset to display. The complex reasoning structures shown in the figure can be further combined to create even more intricate query structures. At the bottom, we present the numerical computation query structures that we defined for the first time.

Table 8: The number and types of queries sampled from the three datasets (1)

| Type | Query Name | Query Structure Definition | FB15K | DB15K | YAGO15K |
|---|---|---|---|---|---|
| 1p | 1np | ('nv', ('np',)) | 1737 | 3894 | 1271 |
| | 1ap | ('e', ('ap',)) | 2306 | 3407 | 2326 |
| | 1rap | ('nv', ('rap',)) | 1059 | 1085 | 1133 |
| | 1p | ('e', ('rp',)) | 19953 | 3029 | 6208 |
| 2p | an | (('e', ('ap',)), ('np',)) | 2735 | 4070 | 1641 |
| | 2p | (('e', ('rp',)), ('rp',)) | 18355 | 2088 | 1273 |
| | nr | (('nv', ('np',)), ('rap',)) | 2329 | 2197 | 1277 |
| | 2n | (('nv', ('np',)), ('np',)) | 2146 | 4655 | 2294 |
| | ra | (('nv', ('rap',)), ('ap',)) | 587 | 2481 | 924 |
| | pa | (('e', ('rp',)), ('ap',)) | 5257 | 4833 | 2988 |
| | ar | (('e', ('ap',)), ('rap',)) | 4570 | 5072 | 3662 |
| | rp | (('nv', ('rap',)), ('rp',)) | 5549 | 2615 | 2470 |
| 2i | ani | (('e', ('ap',)), ('nv', ('np',)), ('i',)) | 7176 | 14727 | 10576 |
| | 2pi | (('e', ('rp',)), ('e', ('rp',)), ('i',)) | 25720 | 1192 | 2574 |
| | nai | (('nv', ('np',)), ('e', ('ap',)), ('i',)) | 7473 | 14829 | 10607 |
| | 2ni | (('nv', ('np',)), ('nv', ('np',)), ('i',)) | 9196 | 13349 | 7765 |
| | 2ai | (('e', ('ap',)), ('e', ('ap',)), ('i',)) | 1976 | 2307 | 3343 |
| | rpi | (('nv', ('rap',)), ('e', ('rp',)), ('i',)) | 5221 | 229 | 132 |
| | pri | (('e', ('rp',)), ('nv', ('rap',)), ('i',)) | 5161 | 214 | 135 |
| | 2ri | (('nv', ('rap',)), ('nv', ('rap',)), ('i',) | 109 | 896 | 61 |
| 3i | nan3i | (('nv', ('np',)), ('e', ('ap',)), ('nv', ('np',)), ('i',)) | 12747 | 21454 | 15296 |
| | 2pr3i | (('e', ('rp',)), ('e', ('rp',)), ('nv', ('rap',)), ('i',)) | 3515 | 19 | 52 |
| | n2a3i | (('nv', ('np',)), ('e', ('ap',)), ('e', ('ap',)), ('i',)) | 3703 | 3735 | 5989 |
| | r2p3i | (('nv', ('rap',)), ('e', ('rp',)), ('e', ('rp',)), ('i',)) | 3527 | 16 | 45 |
| | a2n3i | (('e', ('ap',)), ('nv', ('np',)), ('nv', ('np',)), ('i',)) | 12641 | 21321 | 15287 |
| | 3p3i | (('e', ('rp',)), ('e', ('rp',)), ('e', ('rp',)), ('i',)) | 18859 | 294 | 898 |
| | p2r3i | (('e', ('rp',)), ('nv', ('rap',)), ('nv', ('rap',)), ('i',)) | 93 | 6 | 6 |
| | 2na3i | (('nv', ('np',)), ('nv', ('np',)), ('e', ('ap',)), ('i',)) | 12459 | 21500 | 15285 |
| | 3a3i | (('e', ('ap',)), ('e', ('ap',)), ('e', ('ap',)), ('i',)) | 1977 | 1327 | 2454 |
| | ana3i | (('e', ('ap',)), ('nv', ('np',)), ('e', ('ap',)), ('i',)) | 3625 | 3723 | 5921 |
| | 3n3i | (('nv', ('np',)), ('nv', ('np',)), ('nv', ('np',)), ('i',)) | 11444 | 11950 | 12818 |
| | prp3i | (('e', ('rp',)), ('nv', ('rap',)), ('e', ('rp',)), ('i',)) | 3562 | 24 | 46 |
| | 2an3i | (('e', ('ap',)), ('e', ('ap',)), ('nv', ('np',)), ('i',)) | 3724 | 3716 | 5874 |
| | 2rp3i | (('nv', ('rap',)), ('nv', ('rap',)), ('e', ('rp',)), ('i',)) | 97 | 5 | 4 |
| | rpr3i | (('nv', ('rap',)), ('e', ('rp',)), ('nv', ('rap',)), ('i',)) | 97 | 2 | 7 |
| | 3r3i | (('nv', ('rap',)), ('nv', ('rap',)), ('nv', ('rap',)), ('i',)) | 1 | 17 | 4 |
| ip | 2pip | ((('e', ('rp',)), ('e', ('rp',)), ('i',)), ('rp',)) | 14745 | 783 | 828 |
| | nair | ((('nv', ('np',)), ('e', ('ap',)), ('i',)), ('rap',)) | 4849 | 4748 | 4436 |
| | 2pia | ((('e', ('rp',)), ('e', ('rp',)), ('i',)), ('ap',)) | 7104 | 3468 | 1960 |
| | 2nir | ((('nv', ('np',)), ('nv', ('np',)), ('i',)), ('rap',)) | 8581 | 4588 | 6937 |
| | nain | ((('nv', ('np',)), ('e', ('ap',)), ('i',)), ('np',)) | 5846 | 9170 | 5772 |
| | 2nin | ((('nv', ('np',)), ('nv', ('np',)), ('i',)), ('np',)) | 5064 | 8043 | 6116 |
| | 2air | ((('e', ('ap',)), ('e', ('ap',)), ('i',)), ('rap',)) | 4196 | 4068 | 3189 |
| | rpip | ((('nv', ('rap',)), ('e', ('rp',)), ('i',)), ('rp',)) | 5910 | 754 | 740 |
| | anin | ((('e', ('ap',)), ('nv', ('np',)), ('i',)), ('np',)) | 5906 | 9225 | 5898 |
| | prip | ((('e', ('rp',)), ('nv', ('rap',)), ('i',)), ('rp',)) | 6020 | 814 | 723 |
| | 2rip | ((('nv', ('rap',)), ('nv', ('rap',)), ('i',)), ('rp',)) | 1683 | 996 | 464 |
| | 2ain | ((('e', ('ap',)), ('e', ('ap',)), ('i',)), ('np',)) | 2262 | 2289 | 1981 |
| | anir | ((('e', ('ap',)), ('nv', ('np',)), ('i',)), ('rap',)) | 4851 | 4581 | 4484 |
| | rpia | ((('nv', ('rap',)), ('e', ('rp',)), ('i',)), ('ap',)) | 434 | 157 | 22 |
| | pria | ((('e', ('rp',)), ('nv', ('rap',)), ('i',)), ('ap',)) | 379 | 168 | 23 |
| | 2ria | ((('nv', ('rap',)), ('nv', ('rap',)), ('i',)), ('ap',)) | 11 | 459 | 13 |

Table 9: The number and types of queries sampled from the three datasets (2)

| Type | Query Name | Query Structure Definition | FB15K | DB15K | YAGO15K |
|------|-----------|---------------------------|-------|-------|---------|
| pi | anni | ((('e', ('ap',)), ('np',)), ('nv', ('np',)), ('i',)) | 7960 | 13633 | 9102 |
|  | rppi | ((('nv', ('rap',)), ('rp',)), ('e', ('rp',)), ('i',)) | 6511 | 618 | 1381 |
|  | nrpi | ((('nv', ('np',)), ('rap',)), ('e', ('rp',)), ('i',)) | 13161 | 2829 | 5621 |
|  | paai | ((('e', ('rp',)), ('ap',)), ('e', ('ap',)), ('i',)) | 1333 | 1245 | 1349 |
|  | 2nni | ((('nv', ('np',)), ('np',)), ('nv', ('np',)), ('i',)) | 6975 | 11904 | 11132 |
|  | pani | ((('e', ('rp',)), ('ap',)), ('nv', ('np',)), ('i',)) | 10097 | 10100 | 7629 |
|  | 2nai | ((('nv', ('np',)), ('np',)), ('e', ('ap',)), ('i',)) | 4537 | 11300 | 7658 |
|  | anai | ((('e', ('ap',)), ('np',)), ('e', ('ap',)), ('i',)) | 4468 | 9727 | 7520 |
|  | 2ppi | ((('e', ('rp',)), ('rp',)), ('e', ('rp',)), ('i',)) | 16875 | 541 | 1063 |
|  | nrri | ((('nv', ('np',)), ('rap',)), ('nv', ('rap',)), ('i',)) | 1375 | 3247 | 1291 |
|  | 2pri | ((('e', ('rp',)), ('rp',)), ('nv', ('rap',)), ('i',)) | 5806 | 91 | 64 |
|  | rani | ((('nv', ('rap',)), ('ap',)), ('nv', ('np',)), ('i',)) | 804 | 3258 | 1398 |
|  | arpi | ((('e', ('ap',)), ('rap',)), ('e', ('rp',)), ('i',)) | 2933 | 139 | 131 |
|  | rpri | ((('nv', ('rap',)), ('rp',)), ('nv', ('rap',)), ('i',)) | 1467 | 570 | 112 |
|  | arri | ((('e', ('ap',)), ('rap',)), ('nv', ('rap',)), ('i',)) | 104 | 732 | 73 |
|  | raai | ((('e', ('rap',)), ('ap',)), ('nv', ('ap',)), ('i',)) | 65 | 11 | 42 |
| 2u | pru | (('e', ('rp',)), ('nv', ('rap',)), ('u',)) | 14157 | 5705 | 6783 |
|  | 2pu | (('e', ('rp',)), ('e', ('rp',)), ('u',)) | 19129 | 3734 | 2964 |
|  | anu | (('e', ('ap',)), ('nv', ('np',)), ('u',)) | 12581 | 23056 | 20887 |
|  | 2nu | (('nv', ('np',)), ('nv', ('np',)), ('u',)) | 11988 | 21767 | 11051 |
|  | rpu | (('nv', ('rap',)), ('e', ('rp',)), ('u',)) | 14195 | 5591 | 6641 |
|  | nau | (('nv', ('np',)), ('e', ('ap',)), ('u',)) | 12757 | 23000 | 21086 |
|  | 2ru | (('nv', ('rap',)), ('nv', ('rap',)), ('u',)) | 9359 | 7740 | 14718 |
|  | 2au | (('e', ('ap',)), ('e', ('ap',)), ('u',)) | 2548 | 5497 | 3840 |
| up | 2aur | ((('e', ('ap',)), ('e', ('ap',)), ('u',)), ('rap',)) | 3403 | 2913 | 3922 |
|  | rpup | ((('nv', ('rap',)), ('e', ('rp',)), ('u',)), ('rp',)) | 6871 | 1718 | 1259 |
|  | 2aun | ((('e', ('ap',)), ('e', ('ap',)), ('u',)), ('np',)) | 6705 | 12892 | 10000 |
|  | naur | ((('nv', ('np',)), ('e', ('ap',)), ('u',)), ('rap',)) | 7195 | 5387 | 12506 |
|  | 2pup | ((('e', ('rp',)), ('e', ('rp',)), ('u',)), ('rp',)) | 10171 | 1022 | 574 |
|  | rpua | ((('nv', ('rap',)), ('e', ('rp',)), ('u',)), ('ap',)) | 3494 | 3352 | 1770 |
|  | anun | ((('e', ('ap',)), ('nv', ('np',)), ('u',)), ('np',)) | 7186 | 14110 | 9244 |
|  | 2nur | ((('nv', ('np',)), ('nv', ('np',)), ('u',)), ('rap',)) | 8816 | 6578 | 9445 |
|  | anur | ((('e', ('ap',)), ('nv', ('np',)), ('u',)), ('rap',)) | 7071 | 5473 | 12582 |
|  | 2rup | ((('nv', ('rap',)), ('nv', ('rap',)), ('u',)), ('rp',)) | 4336 | 2610 | 2807 |
|  | naun | ((('nv', ('np',)), ('e', ('ap',)), ('u',)), ('np',)) | 7337 | 14341 | 9114 |
|  | 2pua | ((('e', ('rp',)), ('e', ('rp',)), ('u',)), ('ap',)) | 6981 | 3095 | 1245 |
|  | 2nun | ((('nv', ('np',)), ('nv', ('np',)), ('u',)), ('np',)) | 7575 | 14475 | 8895 |
|  | prup | ((('e', ('rp',)), ('nv', ('rap',)), ('u',)), ('rp',)) | 6779 | 1673 | 1204 |
|  | prua | ((('e', ('rp',)), ('nv', ('rap',)), ('u',)), ('ap',)) | 3631 | 3295 | 1768 |
|  | 2rua | ((('nv', ('rap',)), ('nv', ('rap',)), ('u',)), ('ap',)) | 864 | 4971 | 2133 |
| 2b | aab | (('e', ('ap',)), ('e', ('ap',)), ('b',)) | 1997 | 1997 | 1997 |
| 3b | 3ab | (('e', ('ap',)), ('e', ('ap',)), ('e', ('ap',)), ('b',)) | 1997 | 1997 | 1997 |
| 2abp | aabr | ((('e', ('ap',)), ('e', ('ap',)), ('b',)), ('rap',)) | 885 | 431 | 1690 |
|  | aabn | ((('e', ('ap',)), ('e', ('ap',)), ('b',)), ('np',)) | 1110 | 1565 | 305 |
| p2ab | raab | ((('nv', ('rap',)), ('ap',)), ('e', ('ap',)), ('b',)) | 349 | 862 | 1225 |
|  | paab | ((('e', ('rp',)), ('ap',)), ('e', ('ap',)), ('b',)) | 1647 | 1134 | 771 |
| 2p2ab | rarab | ((('nv', ('rap',)), ('ap',)), (('e', ('rp',)), ('ap',)), ('b',)) | 510 | 365 | 826 |
|  | rapab | ((('nv', ('rap',)), ('ap',)), (('nv', ('rap',)), ('ap',)), ('b',)) | 99 | 504 | 481 |
|  | parab | ((('e', ('rp',)), ('ap',)), (('nv', ('rap',)), ('ap',)), ('b',)) | 519 | 517 | 457 |
|  | papab | ((('e', ('rp',)), ('ap',)), (('e', ('rp',)), ('ap',)), ('b',)) | 866 | 608 | 230 |

Table 10: Percentage of edges on the inference path that exist in the Test Set.

| Dataset | AVG | 2p | 2i | 3i | pi | ip | 2u | up |
|---------|-----|-----|-----|-----|-----|-----|-----|-----|
| FB15K | 0.7537 | 0.6505 | 0.9123 | 0.9007 | 0.7506 | 0.5644 | 0.7364 | 0.7612 |
| DB15K | 0.6298 | 0.5488 | 0.6892 | 0.5695 | 0.5483 | 0.6915 | 0.6178 | 0.7433 |
| YAGO15K | 0.6048 | 0.5627 | 0.7651 | 0.6628 | 0.6924 | 0.5106 | 0.6272 | 0.4128 |

## C  MODEL HYPERPARAMETER SETTINGS AND TRAINING DETAILS

The hyperparameter settings for CNR-NST are shown in Table 9. $W_{rel}$ refers to the score weight of the relation, $W_{EAV}$ refers to the task score weight in joint training, and Thrshd refers to the threshold of the corresponding adjacency matrix. Thrshd$_{fuzzy}$ refers to the filtering value for numerical fuzzy sets during binary operations, which is used to speed up the inference process.

Table 11: Hyperparameter Settings of CNR-NST

| | Training Epoch | Learning Rate | Ranking | Decay1 | Decay2 | $W_{rel}$ | $W_{EAV}$ | $W_{VFV}$ |
|---|---|---|---|---|---|---|---|---|
| Multi-ComplEx | 100 | 0.05 | 500 | 0.9 | 0.999 | 4 | 5 | 0.1 |

| | Fraction | Thrshd$_{ERE}$ | Thrshd$_{EAV}$ | Thrshd$_{VFV}$ | Thrshd$_{fuzzy}$ | Neg$_{Scale}$ | | |
|---|---|---|---|---|---|---|---|---|
| Reasoning-Model | 10 | 0.001 | 0.0001 | 0.0001 | 0.001 | 6 | | |

## D  DETAILED EXPERIMENTAL RESULTS

### D.1  MAIN EXPERIMENTAL RESULTS

Here, we present the MRR results for all 102 sub-queries on our dataset. We applied two sampling methods on the public datasets FB15K, DB15K, and YAGO15K, resulting in 77 query types using the NRN sampling method and 92 query types using our improved method. Additionally, we provide detailed results for the 10 extended numerical computation query types.

The detailed data shows that, while Q2B+NRN and GQE+NRN exhibit decent performance on some sub-tasks involving intersection, they perform poorly on other tasks. Q2P+NRN demonstrates better overall performance, but our CNR-NST model significantly outperforms Q2P+NRN in both average performance and numerical query tasks (see Section 4.2 of the main text for average performance).

The definition of query subclass names: "p" stands for "Relation Prediction", "a" stands for "Attribute Prediction", "r" stands for "Reverse Attribute Prediction", "n" stands for "Numerical Prediction", "u" stands for "Union", "i" stands for "Intersection", and "b" stands for "Binary Operator".

Table 12: Detailed MRR Results (%) in the FB15K Test Set

| Query_num | Method | 1rap | 1rp | 1ap | 1np | pa | ar | rp | ra | an | 2n | 2p | nr |
|---|---|---|---|---|---|---|---|---|---|---|---|---|---|
| 77 | GQE+NRN | 1.62 | 30.64 | - | 1.31 | 9.30 | 0.25 | 12.40 | 1.12 | 1.07 | 2.29 | 6.81 | 0.71 |
| | Q2B+NRN | 2.16 | 39.56 | - | 1.26 | 9.66 | 0.26 | 8.89 | 0.70 | 0.99 | 2.67 | 7.09 | 0.71 |
| | Q2P+NRN | 1.79 | 40.95 | - | 0.06 | 13.72 | 0.64 | 11.07 | 1.01 | 0.10 | 0.09 | 7.80 | 0.70 |
| | CNR-NST | 6.13 | 66.92 | - | 30.15 | 21.52 | 2.64 | 40.93 | 4.43 | 2.04 | 2.69 | 47.53 | 2.29 |
| 92 | GQE+NRN | 1.36 | 31.18 | 5.22 | 1.41 | 10.91 | 0.25 | 13.23 | 1.32 | 2.04 | 1.61 | 6.69 | 0.72 |
| | Q2B+NRN | 2.21 | 39.25 | 5.14 | 1.57 | 11.17 | 0.35 | 8.76 | 0.62 | 2.30 | 2.42 | 6.84 | 0.81 |
| | Q2P+NRN | 0.26 | 37.52 | 0.78 | 0.05 | 2.97 | 0.07 | 13.01 | 0.30 | 0.12 | 0.06 | 10.75 | 0.44 |
| | CNR-NST | 6.13 | 66.92 | 4.21 | 31.31 | 25.81 | 2.61 | 46.89 | 1.61 | 17.46 | 14.49 | 46.16 | 0.80 |

| Query_num | Method | 2pi | 2ai | pri | rpi | 2ri | 2ni | nai | ani |
|---|---|---|---|---|---|---|---|---|---|
| 77 | GQE+NRN | 27.65 | - | 29.97 | 30.11 | 4.80 | 1.70 | | |
| | Q2B+NRN | 31.41 | - | 31.18 | 31.41 | 9.64 | 1.77 | | |
| | Q2P+NRN | 37.22 | - | 35.73 | 36.05 | 25.85 | 0.02 | | |
| | CNR-NST | 56.21 | - | 42.44 | 41.69 | 6.53 | 2.30 | | |
| 92 | GQE+NRN | 27.83 | 58.17 | 30.11 | 30.70 | 3.61 | 1.30 | 35.64 | 35.27 |
| | Q2B+NRN | 31.29 | 52.34 | 31.17 | 31.75 | 6.52 | 1.44 | 35.91 | 35.22 |
| | Q2P+NRN | 33.30 | 22.45 | 30.89 | 30.50 | 0.98 | 0.21 | 10.76 | 10.91 |
| | CNR-NST | 56.06 | 43.94 | 35.16 | 35.47 | 5.32 | 8.93 | 23.52 | 20.61 |

| Query_num | Method | 2na3i | 3n3i | r2p3i | rpr3i | nan3i | n2a3i | 2rp3i | 3r3i | prp3i | p2r3i | 3p3i | 2pr3i | 2an3i | 3a3i | a2n3i | ana3i |
|---|---|---|---|---|---|---|---|---|---|---|---|---|---|---|---|---|---|
| 77 | GQE+NRN | - | 1.98 | 41.98 | 38.00 | - | - | 37.26 | - | 41.96 | 41.90 | 32.65 | 41.55 | - | - | - | - |
| | Q2B+NRN | - | 2.31 | 43.18 | 39.67 | - | - | 34.31 | - | 43.35 | 34.45 | 35.37 | 42.25 | - | - | - | - |
| | Q2P+NRN | - | 0.03 | 47.59 | 56.41 | - | - | 47.67 | - | 47.54 | 42.33 | 39.81 | 47.54 | - | - | - | - |
| | CNR-NST | - | 5.89 | 54.02 | 42.10 | - | - | 38.90 | - | 54.30 | 41.05 | 54.02 | 54.24 | - | - | - | - |
| 92 | GQE+NRN | 49.98 | 1.74 | 42.07 | 29.54 | 49.99 | 83.45 | 40.36 | 0.30 | 41.67 | 41.54 | 32.90 | 42.67 | 49.53 | 87.23 | 49.99 | 84.09 |
| | Q2B+NRN | 50.00 | 1.68 | 43.23 | 37.22 | 49.40 | 84.72 | 27.39 | 0.48 | 42.58 | 39.52 | 35.27 | 42.27 | 85.77 | 95.19 | 49.76 | 84.70 |
| | Q2P+NRN | 15.83 | 0.22 | 43.53 | 36.49 | 15.70 | 24.81 | 28.90 | 7.14 | 42.92 | 39.83 | 37.79 | 43.99 | 26.90 | 15.88 | 15.76 | 26.93 |
| | CNR-NST | 15.21 | 3.64 | 42.54 | 41.10 | 14.86 | 25.08 | 46.81 | 2.63 | 42.12 | 46.13 | 55.41 | 43.85 | 27.14 | 49.98 | 14.88 | 25.79 |

| Query_num | Method | pria | aair | ppia | ppip | prip | rpip | rrip | rpia | aain | nain | nair | anin | nnin | nnir | anir | rria |
|---|---|---|---|---|---|---|---|---|---|---|---|---|---|---|---|---|---|
| 77 | GQE+NRN | 26.90 | 0.18 | 13.31 | 11.80 | 12.33 | 12.77 | 11.07 | 26.38 | 1.05 | 0.75 | 1.22 | 0.62 | 1.67 | 1.03 | 0.21 | 0.04 |
| | Q2B+NRN | 26.66 | 0.19 | 15.34 | 12.49 | 12.94 | 13.28 | 14.80 | 27.87 | 1.34 | 0.80 | 0.26 | 0.66 | 2.14 | 0.98 | 0.26 | 0.05 |
| | Q2P+NRN | 17.44 | 0.34 | 17.97 | 13.58 | 16.71 | 16.02 | 18.97 | 15.31 | 0.08 | 0.07 | 0.61 | 0.07 | 0.09 | 0.32 | 0.62 | 100.00 |
| | CNR-NST | 35.80 | 8.96 | 27.60 | 50.38 | 55.06 | 55.14 | 57.86 | 35.13 | 2.45 | 1.34 | 3.35 | 1.28 | 0.98 | 0.71 | 3.28 | 5.04 |
| 92 | GQE+NRN | 26.80 | 0.16 | 18.85 | 12.13 | 12.41 | 12.58 | 10.24 | 24.98 | 5.64 | 6.13 | 0.20 | 5.97 | 1.48 | 0.98 | 0.17 | 0.04 |
| | Q2B+NRN | 24.23 | 0.14 | 20.20 | 12.48 | 12.98 | 13.55 | 14.60 | 27.47 | 9.29 | 7.30 | 0.24 | 7.44 | 1.57 | 0.93 | 7.30 | 0.06 |
| | Q2P+NRN | 8.36 | 0.08 | 6.34 | 14.60 | 15.71 | 15.21 | 15.64 | 7.94 | 0.72 | 0.97 | 0.07 | 0.90 | 0.09 | 0.77 | 0.06 | 0.03 |
| | CNR-NST | 34.17 | 8.96 | 33.09 | 50.38 | 54.77 | 55.57 | 62.94 | 31.61 | 12.59 | 8.03 | 2.98 | 8.22 | 4.58 | 0.58 | 1.23 | 0.43 |

| Query_num | Method | pppi | ppri | arpi | rppi | arri | rpri | nnai | pani | nnni | anni | anai | rani | nrpi | raai | nrri | paai |
|---|---|---|---|---|---|---|---|---|---|---|---|---|---|---|---|---|---|
| 77 | GQE+NRN | 17.71 | 19.05 | 26.19 | 21.09 | 2.58 | 21.33 | - | 9.89 | 1.97 | 1.00 | - | 2.31 | 10.80 | - | 3.71 | - |
| | Q2B+NRN | 22.02 | 19.31 | 27.32 | 23.15 | 5.30 | 20.02 | - | 11.51 | 2.59 | 1.23 | - | 2.85 | 13.09 | - | 4.44 | - |
| | Q2P+NRN | 24.88 | 22.35 | 36.27 | 47.54 | 13.85 | 28.54 | - | 18.28 | 0.03 | 0.02 | - | 3.02 | 19.05 | - | 26.01 | - |
| | CNR-NST | 50.71 | 33.50 | 40.23 | 51.24 | 21.40 | 42.30 | - | 8.83 | 1.44 | 2.05 | - | 15.24 | 12.25 | - | 6.44 | - |
| 92 | GQE+NRN | 17.70 | 19.55 | 25.40 | 21.37 | 2.22 | 21.24 | 39.87 | 15.34 | 1.65 | 7.43 | 41.95 | 2.60 | 11.08 | 5.79 | 2.72 | 50.67 |
| | Q2B+NRN | 21.18 | 20.63 | 26.84 | 23.03 | 4.38 | 21.07 | 40.35 | 16.93 | 1.95 | 7.27 | 42.54 | 1.69 | 12.78 | 5.92 | 4.34 | 51.96 |
| | Q2P+NRN | 23.36 | 20.94 | 26.58 | 23.88 | 1.17 | 21.56 | 16.41 | 5.26 | 0.28 | 1.27 | 14.07 | 0.28 | 12.08 | 3.26 | 2.67 | 18.31 |
| | CNR-NST | 51.14 | 29.16 | 32.62 | 50.34 | 4.01 | 32.43 | 37.24 | 7.39 | 7.23 | 8.57 | 39.58 | 2.05 | 13.85 | 48.42 | 2.63 | 56.79 |

| Query_num | Method | 2pu | rpu | 2ru | pru | 2au | 2nu | anu | nau |
|---|---|---|---|---|---|---|---|---|---|
| 77 | GQE+NRN | 7.97 | 8.14 | 0.55 | 7.97 | 1.19 | 0.52 | 0.48 | 0.46 |
| | Q2B+NRN | 12.65 | 11.95 | 0.35 | 11.85 | 1.31 | 0.62 | 0.74 | 0.67 |
| | Q2P+NRN | 19.43 | 19.87 | 0.60 | 18.98 | 0.45 | 0.11 | 0.08 | 0.09 |
| | CNR-NST | 45.52 | 18.03 | 1.12 | 18.03 | 4.10 | 1.01 | 2.77 | 2.75 |
| 92 | GQE+NRN | 7.91 | 7.73 | 0.57 | 7.77 | 0.96 | 0.55 | 0.54 | 0.52 |
| | Q2B+NRN | 12.66 | 11.87 | 0.42 | 11.78 | 1.17 | 0.67 | 0.82 | 0.77 |
| | Q2P+NRN | 14.72 | 14.74 | 0.09 | 15.10 | 0.11 | 0.16 | 0.15 | 0.15 |
| | CNR-NST | 44.91 | 36.30 | 2.02 | 35.89 | 1.19 | 4.15 | 7.40 | 7.22 |

| Query_num | Method | 2pup | 2pua | prup | prua | anun | anur | 2aur | 2aun | naur | naun | 2nun | 2nur | 2rup | 2rua | rpua | rpup |
|---|---|---|---|---|---|---|---|---|---|---|---|---|---|---|---|---|---|
| 77 | GQE+NRN | 5.55 | 5.26 | 5.71 | 3.77 | 0.83 | 0.59 | 0.87 | 0.33 | 0.65 | 0.83 | 0.72 | 0.75 | 7.11 | 0.94 | 3.83 | 5.92 |
| | Q2B+NRN | 5.87 | 6.47 | 6.12 | 5.45 | 1.10 | 0.54 | 0.70 | 0.45 | 0.58 | 1.02 | 0.89 | 0.65 | 6.92 | 1.02 | 5.56 | 6.10 |
| | Q2P+NRN | 6.03 | 10.75 | 7.55 | 9.31 | 0.09 | 0.62 | 3.39 | 0.10 | 0.62 | 0.08 | 0.08 | 0.58 | 8.15 | 1.13 | 8.05 | 6.91 |
| | CNR-NST | 44.62 | 9.31 | 39.35 | 9.72 | 0.97 | 1.46 | 2.43 | 0.91 | 1.47 | 0.95 | 0.85 | 1.12 | 31.03 | 1.22 | 10.17 | 39.58 |
| 92 | GQE+NRN | 5.51 | 5.97 | 5.67 | 4.22 | 0.99 | 0.65 | 1.05 | 0.71 | 0.67 | 0.96 | 0.64 | 0.76 | 6.76 | 0.92 | 4.22 | 5.86 |
| | Q2B+NRN | 5.51 | 6.89 | 6.05 | 5.00 | 1.64 | 0.54 | 0.71 | 1.05 | 0.62 | 1.58 | 0.88 | 0.68 | 6.79 | 1.09 | 5.11 | 6.09 |
| | Q2P+NRN | 9.17 | 1.00 | 8.72 | 0.88 | 0.18 | 0.36 | 0.75 | 0.15 | 0.41 | 0.22 | 0.15 | 0.46 | 8.78 | 0.39 | 0.79 | 8.87 |
| | CNR-NST | 43.78 | 22.10 | 42.42 | 21.13 | 3.25 | 0.95 | 2.37 | 5.61 | 0.95 | 3.53 | 2.33 | 0.73 | 40.73 | 1.46 | 19.89 | 42.36 |

| Query_num | Method | 2b | 3b | bp | | pb | | 2pb | | | |
|---|---|---|---|---|---|---|---|---|---|---|---|
| | | aab | aaab | aabr | aabn | raab | paab | rarab | rapab | parab | papab |
| 102 | CNR-NST | 18.39 | 17.56 | 4.08 | 5.64 | 16.62 | 28.24 | 7.23 | 14.68 | 15.53 | 26.93 |

Table 13: Detailed MRR Results (%) in the DB15K Test Set

| Query_num | Method | 1rap | 1rp | 1ap | 1np | pa | ar | rp | ra | an | 2n | 2p | nr |
|---|---|---|---|---|---|---|---|---|---|---|---|---|---|
| 77 | GQE+NRN | 1.51 | 8.06 | 3.65 | 0.77 | 4.28 | 0.25 | 6.55 | 1.38 | 0.65 | 0.66 | 7.45 | 0.51 |
|  | Q2B+NRN | 2.17 | 9.67 | 3.80 | 0.97 | 4.42 | 0.42 | 5.50 | 1.30 | 0.71 | 0.75 | 8.50 | 0.60 |
|  | Q2P+NRN | 0.72 | 17.29 | 0.86 | 0.02 | 1.47 | 0.18 | 8.18 | 0.37 | 0.02 | 0.02 | 13.40 | 0.47 |
|  | CNR-NST | 4.61 | 26.73 | 4.61 | 7.76 | 9.63 | 2.20 | 16.30 | 5.30 | 6.66 | 2.83 | 24.06 | 0.35 |
| 92 | GQE+NRN | 1.52 | 8.04 | 2.96 | 0.82 | 6.00 | 0.22 | 6.15 | 1.54 | 1.74 | 0.67 | 7.12 | 0.47 |
|  | Q2B+NRN | 2.30 | 10.10 | 2.25 | 1.05 | 7.09 | 0.33 | 5.19 | 1.25 | 1.95 | 0.95 | 8.39 | 0.56 |
|  | Q2P+NRN | 0.76 | 17.67 | 0.41 | 0.03 | 1.56 | 0.17 | 7.86 | 0.34 | 0.04 | 0.02 | 13.60 | 0.36 |
|  | CNR-NST | 4.31 | 26.64 | 4.25 | 9.29 | 9.60 | 2.20 | 15.55 | 2.59 | 6.46 | 5.09 | 25.99 | 1.04 |

| Query_num | Method | 2pi | 2ai | pri | rpi | 2ri | 2ni | nai | ani |
|---|---|---|---|---|---|---|---|---|---|
| 77 | GQE+NRN | 30.74 | - | 29.41 | 29.05 | 14.36 | 1.20 | 8.76 | 9.66 |
|  | Q2B+NRN | 28.64 | - | 33.12 | 33.26 | 18.30 | 1.41 | 8.67 | 10.46 |
|  | Q2P+NRN | 41.69 | - | 37.64 | 38.61 | 15.31 | 0.04 | 7.89 | 6.93 |
|  | CNR-NST | 46.16 | - | 35.58 | 34.66 | 18.52 | 2.52 | 12.99 | 12.23 |
| 92 | GQE+NRN | 31.19 | 36.46 | 25.24 | 26.99 | 12.55 | 1.03 | 27.14 | 27.17 |
|  | Q2B+NRN | 31.20 | 36.76 | 28.02 | 29.10 | 17.31 | 1.24 | 28.89 | 28.78 |
|  | Q2P+NRN | 40.81 | 21.36 | 33.68 | 31.75 | 15.71 | 0.06 | 2.50 | 2.48 |
|  | CNR-NST | 48.62 | 35.68 | 33.40 | 32.43 | 18.87 | 3.37 | 16.03 | 14.03 |

| Query_num | Method | 2na3i | 3n3i | r2p3i | rpr3i | nan3i | n2a3i | 2rp3i | 3r3i | prp3i | p2r3i | 3p3i | 2pr3i | 2an3i | 3a3i | a2n3i | ana3i |
|---|---|---|---|---|---|---|---|---|---|---|---|---|---|---|---|---|---|
| 77 | GQE+NRN | 12.66 | 1.33 | 70.37 | 70.50 | 11.09 | - | 73.58 | 32.12 | 60.82 | 61.25 | 44.41 | 70.38 | - | - | 11.12 | - |
|  | Q2B+NRN | 16.08 | 1.56 | 69.75 | 84.03 | 13.26 | - | 73.88 | 32.54 | 63.88 | 93.86 | 43.94 | 66.23 | - | - | 15.57 | - |
|  | Q2P+NRN | 11.20 | 0.09 | 79.08 | 83.70 | 9.88 | - | 73.70 | 29.73 | 82.65 | 81.36 | 55.69 | 81.74 | - | - | 11.55 | - |
|  | CNR-NST | 8.81 | 1.15 | 65.61 | 71.06 | 8.74 | 13.08 | 67.05 | 42.71 | 64.54 | 76.68 | 73.33 | 59.05 | 12.87 | 39.38 | 8.54 | 12.10 |
| 92 | GQE+NRN | 49.27 | 1.24 | 73.53 | 69.44 | 48.77 | 76.02 | 43.75 | 35.19 | 79.03 | 51.04 | 47.87 | 75.72 | 76.76 | 83.67 | 47.81 | 76.17 |
|  | Q2B+NRN | 48.14 | 1.36 | 64.60 | 57.27 | 47.63 | 77.07 | 51.71 | 29.33 | 72.84 | 60.49 | 45.08 | 73.19 | 77.28 | 85.53 | 46.90 | 77.41 |
|  | Q2P+NRN | 3.26 | 0.18 | 70.57 | 73.61 | 3.26 | 17.06 | 73.61 | 26.28 | 76.00 | 70.83 | 53.60 | 53.89 | 16.92 | 19.03 | 3.17 | 19.12 |
|  | CNR-NST | 15.56 | 1.84 | 72.57 | 100.00 | 21.37 | 25.74 | 80.00 | 31.60 | 76.39 | 53.89 | 61.51 | 72.11 | 31.33 | 57.06 | 21.18 | 39.06 |

| Query_num | Method | pria | aair | ppia | ppip | prip | rpip | rrip | rpia | aain | nain | nair | anin | nnin | nnir | anir | rria |
|---|---|---|---|---|---|---|---|---|---|---|---|---|---|---|---|---|---|
| 77 | GQE+NRN | 10.19 | 0.32 | 15.59 | 11.29 | 11.45 | 11.84 | 6.62 | 10.19 | 1.05 | 0.30 | 0.23 | 0.32 | 0.66 | 0.62 | 0.28 | 7.61 |
|  | Q2B+NRN | 13.17 | 0.46 | 17.12 | 11.49 | 12.04 | 13.53 | 7.29 | 12.82 | 1.16 | 0.34 | 0.41 | 0.29 | 0.77 | 0.50 | 0.31 | 9.32 |
|  | Q2P+NRN | 3.81 | 0.18 | 6.17 | 18.16 | 18.52 | 18.89 | 9.18 | 5.65 | 0.04 | 0.03 | 0.14 | 0.03 | 0.04 | 0.41 | 0.12 | 2.01 |
|  | CNR-NST | 11.14 | 2.18 | 28.31 | 28.65 | 26.75 | 27.89 | 16.85 | 16.86 | 1.73 | 2.15 | 0.70 | 2.42 | 1.22 | 0.25 | 0.57 | 9.84 |
| 92 | GQE+NRN | 12.13 | 0.29 | 22.46 | 12.25 | 11.72 | 11.74 | 6.84 | 12.85 | 5.26 | 4.87 | 0.24 | 4.95 | 0.62 | 0.46 | 0.18 | 7.09 |
|  | Q2B+NRN | 12.64 | 0.33 | 29.17 | 13.68 | 11.71 | 11.77 | 7.73 | 12.64 | 6.88 | 6.79 | 0.32 | 6.89 | 0.73 | 0.45 | 0.27 | 8.16 |
|  | Q2P+NRN | 4.57 | 0.14 | 6.41 | 16.97 | 15.92 | 17.66 | 9.14 | 2.96 | 0.14 | 0.11 | 0.12 | 0.13 | 0.03 | 0.39 | 0.16 | 2.61 |
|  | CNR-NST | 14.57 | 1.30 | 16.35 | 32.96 | 28.49 | 29.33 | 14.84 | 17.39 | 4.32 | 6.94 | 1.21 | 8.46 | 3.94 | 1.49 | 0.96 | 9.44 |

| Query_num | Method | pppi | ppri | arpi | rppi | arri | rpri | nnai | pani | nnni | anni | anai | rani | nrpi | raai | nrri | paai |
|---|---|---|---|---|---|---|---|---|---|---|---|---|---|---|---|---|---|
| 77 | GQE+NRN | 24.64 | 22.52 | 18.03 | 26.02 | - | 8.88 | 14.33 | 5.86 | 5.54 | 1.38 | 0.69 | 1.98 | 3.47 | 23.83 | 0.59 | 11.40 |
|  | Q2B+NRN | 22.92 | 23.00 | 22.94 | 26.46 | - | 11.65 | 14.85 | 5.29 | 4.60 | 1.61 | 0.92 | 2.24 | 2.79 | 31.63 | 1.60 | 15.47 |
|  | Q2P+NRN | 34.79 | 26.99 | 26.95 | 32.48 | - | 10.24 | 14.19 | 3.56 | 2.38 | 0.09 | 0.06 | 2.18 | 0.71 | 37.74 | 1.62 | 13.11 |
|  | CNR-NST | 46.00 | 42.46 | 42.56 | 39.57 | - | 14.07 | 20.15 | 5.58 | 4.15 | 2.28 | 4.15 | 2.29 | 2.59 | 1.70 | 17.11 | 9.39 | 7.71 |
| 92 | GQE+NRN | 25.25 | 24.84 | 22.14 | 24.99 | 44.61 | 8.89 | 11.83 | 40.63 | 14.33 | 1.39 | 9.93 | 38.00 | 3.44 | 23.74 | 0.28 | 11.84 |
|  | Q2B+NRN | 23.15 | 23.42 | 30.72 | 28.44 | 49.46 | 12.81 | 13.15 | 38.36 | 16.66 | 1.71 | 10.37 | 37.57 | 2.70 | 30.42 | 0.08 | 14.39 |
|  | Q2P+NRN | 32.45 | 26.56 | 32.86 | 32.93 | 17.52 | 11.28 | 14.34 | 2.68 | 2.10 | 0.11 | 0.22 | 3.01 | 0.66 | 35.01 | 0.01 | 11.96 |
|  | CNR-NST | 44.18 | 46.01 | 35.98 | 37.21 | 48.90 | 16.11 | 21.66 | 47.60 | 16.59 | 4.53 | 6.71 | 37.24 | 5.78 | 14.90 | 0.92 | 9.40 |

| Query_num | Method | 2pu | rpu | 2ru | pru | 2au | 2nu | anu | nau |
|---|---|---|---|---|---|---|---|---|---|
| 77 | GQE+NRN | 5.40 | 2.03 | 0.45 | 2.22 | 0.29 | 0.21 | 0.27 | 0.30 |
|  | Q2B+NRN | 4.72 | 2.30 | 0.55 | 1.97 | 0.32 | 0.26 | 0.36 | 0.40 |
|  | Q2P+NRN | 12.57 | 3.43 | 0.37 | 3.45 | 0.05 | 0.07 | 0.07 | 0.08 |
|  | CNR-NST | 14.95 | 6.49 | 1.95 | 6.34 | 0.68 | 1.15 | 2.02 | 2.11 |
| 92 | GQE+NRN | 5.27 | 2.25 | 0.56 | 2.07 | 0.32 | 0.22 | 0.18 | 0.21 |
|  | Q2B+NRN | 4.90 | 2.15 | 0.52 | 2.44 | 0.28 | 0.29 | 0.36 | 0.32 |
|  | Q2P+NRN | 11.65 | 3.68 | 0.39 | 3.72 | 0.08 | 0.06 | 0.05 | 0.05 |
|  | CNR-NST | 15.96 | 7.50 | 1.92 | 6.95 | 0.35 | 1.20 | 2.93 | 26.76 |

| Query_num | Method | 2pup | 2pua | prup | prua | anun | anur | 2aur | 2aun | naur | naun | 2nun | 2nur | 2rup | 2rua | rpua | rpup |
|---|---|---|---|---|---|---|---|---|---|---|---|---|---|---|---|---|---|
| 77 | GQE+NRN | 7.62 | 2.56 | 4.37 | 1.53 | 0.39 | 0.72 | 1.64 | 0.18 | 0.86 | 0.40 | 0.33 | 0.52 | - | 1.06 | 1.59 | 4.92 |
|  | Q2B+NRN | 6.25 | 3.24 | 4.90 | 1.47 | 0.44 | 0.71 | 1.58 | 0.21 | 0.60 | 0.42 | 0.37 | 0.57 | 3.80 | 1.16 | 1.65 | 4.28 |
|  | Q2P+NRN | 12.00 | 0.93 | 7.50 | 0.62 | 0.06 | 0.42 | 1.59 | 0.05 | 0.50 | 0.06 | 0.07 | 0.39 | 5.82 | 0.62 | 0.50 | 7.45 |
|  | CNR-NST | 22.22 | 8.44 | 14.56 | 5.12 | 0.91 | 0.80 | 1.87 | 1.56 | 0.96 | 0.92 | 0.51 | 0.36 | 10.53 | 2.95 | 4.74 | 15.84 |
| 92 | GQE+NRN | 6.82 | 3.31 | 4.48 | 1.43 | 0.46 | 0.63 | 1.62 | 0.58 | 0.78 | 0.45 | 0.36 | 0.48 | 3.28 | 0.92 | 1.69 | 4.23 |
|  | Q2B+NRN | 6.64 | 4.62 | 4.65 | 1.41 | 0.64 | 0.55 | 1.51 | 1.07 | 0.59 | 0.67 | 0.42 | 0.51 | 3.71 | 1.15 | 1.68 | 5.24 |
|  | Q2P+NRN | 12.02 | 0.61 | 7.12 | 0.42 | 0.07 | 0.42 | 0.38 | 0.06 | 0.49 | 0.07 | 0.08 | 0.38 | 4.83 | 0.63 | 0.52 | 7.06 |
|  | CNR-NST | 21.97 | 9.25 | 16.82 | 5.17 | 2.47 | 0.88 | 2.23 | 30.06 | 0.95 | 1.84 | 1.12 | 0.58 | 12.23 | 2.23 | 4.43 | 14.77 |

| Query_num | Method | 2b | 3b | bp | | pb | | | | 2pb | |
|---|---|---|---|---|---|---|---|---|---|---|---|
|  |  | aab | aaab | aabr | aabn | raab | paab | rarab | rapab | parab | papab |
| 102 | CNR-NST | 18.39 | 17.56 | 4.08 | 5.64 | 16.62 | 28.24 | 7.23 | 14.68 | 15.53 | 26.93 |

Table 14: Detailed MRR Results (%) in the YAGO15K Test Set

| Query_num | Method | 1rap | 1rp | 1ap | 1np | pa | ar | rp | ra | an | 2n | 2p | nr |
|---|---|---|---|---|---|---|---|---|---|---|---|---|---|
| 77 | GQE+NRN | 1.48 | 5.35 | - | 1.32 | 6.81 | 0.48 | 7.81 | 1.88 | 0.24 | 0.35 | 7.65 | 0.34 |
| | Q2B+NRN | 1.63 | 9.37 | - | 1.41 | 6.87 | 0.61 | 6.53 | 1.72 | 0.28 | 0.35 | 8.26 | 0.23 |
| | Q2P+NRN | 0.12 | 26.72 | - | 0.01 | 10.58 | 0.02 | 14.06 | 0.46 | 0.02 | 0.01 | 3.49 | 0.05 |
| | CNR-NST | 2.96 | 34.60 | - | 21.10 | 13.27 | 1.71 | 18.40 | 4.56 | 19.41 | 7.66 | 22.39 | 1.01 |
| 92 | GQE+NRN | 1.18 | 5.85 | 1.83 | 1.49 | 5.87 | 0.37 | 8.21 | 1.93 | 1.97 | 0.35 | 8.78 | 0.34 |
| | Q2B+NRN | 1.53 | 8.76 | 1.48 | 1.04 | 5.16 | 0.41 | 6.21 | 1.30 | 2.27 | 0.34 | 8.28 | 0.26 |
| | Q2P+NRN | 0.56 | 23.25 | 0.65 | 0.37 | 3.24 | 0.35 | 9.18 | 0.57 | 0.71 | 0.16 | 15.22 | 0.25 |
| | CNR-NST | 2.71 | 34.65 | 2.42 | 22.21 | 13.40 | 1.83 | 17.23 | 5.02 | 21.62 | 7.14 | 22.58 | 1.10 |

| Query_num | Method | 2pi | 2ai | pri | rpi | 2ri | 2ni | nai | ani |
|---|---|---|---|---|---|---|---|---|---|
| 77 | GQE+NRN | 29.89 | - | 34.94 | 37.07 | 30.80 | 0.90 | - | - |
| | Q2B+NRN | 25.69 | - | 33.35 | 32.55 | 39.12 | 0.77 | - | - |
| | Q2P+NRN | 35.52 | - | 58.84 | 56.40 | 30.96 | 0.01 | - | - |
| | CNR-NST | 46.65 | - | 56.10 | 51.93 | 63.00 | 10.20 | - | - |
| 92 | GQE+NRN | 29.92 | 60.58 | 30.57 | 32.34 | 31.01 | 0.62 | 32.59 | 31.44 |
| | Q2B+NRN | 24.91 | 67.49 | 28.86 | 35.57 | 37.78 | 0.83 | 33.54 | 31.21 |
| | Q2P+NRN | 39.24 | 39.37 | 34.80 | 39.88 | 31.23 | 0.29 | 21.16 | 20.42 |
| | CNR-NST | 42.30 | 45.42 | 50.36 | 53.06 | 63.52 | 5.91 | 24.64 | 24.69 |

| Query_num | Method | 2na3i | 3n3i | r2p3i | rpr3i | nan3i | n2a3i | 2rp3i | 3r3i | prp3i | p2r3i | 3p3i | 2pr3i | 2an3i | 3a3i | a2n3i | ana3i |
|---|---|---|---|---|---|---|---|---|---|---|---|---|---|---|---|---|---|
| 77 | GQE+NRN | - | 0.84 | 57.55 | 83.64 | - | - | 88.75 | 100.00 | 53.10 | 85.00 | 45.20 | 50.79 | | | | |
| | Q2B+NRN | - | 0.82 | 53.73 | 84.85 | - | - | 89.58 | 100.00 | 55.80 | 86.11 | 40.52 | 51.10 | | | | |
| | Q2P+NRN | - | 0.01 | 80.39 | 77.47 | - | - | 100.00 | 50.00 | 78.75 | 100.00 | 40.03 | 100.00 | | | | |
| | CNR-NST | - | 2.48 | 70.65 | 83.33 | 46.18 | - | 88.89 | 100.00 | 76.04 | 87.18 | 55.48 | 71.13 | | | | |
| 92 | GQE+NRN | 46.21 | 0.63 | 65.17 | 89.29 | 46.18 | 80.63 | 87.50 | 20.00 | 69.88 | 100.00 | 45.61 | 67.49 | 80.70 | 77.32 | 45.81 | 82.00 |
| | Q2B+NRN | 47.56 | 0.83 | 65.45 | 80.36 | 45.73 | 81.23 | 70.83 | 100.00 | 61.02 | 100.00 | 40.68 | 63.45 | 80.25 | 85.74 | 45.88 | 79.57 |
| | Q2P+NRN | 32.37 | 3.21 | 63.70 | 73.81 | 32.29 | 46.26 | 61.11 | 100.00 | 65.86 | 80.00 | 55.13 | 64.96 | 46.69 | 48.73 | 31.91 | 48.12 |
| | CNR-NST | 21.41 | 6.62 | 76.67 | 92.86 | 22.50 | 48.66 | 70.83 | 100.00 | 80.44 | 72.22 | 53.36 | 70.44 | 48.43 | 55.60 | 21.40 | 42.10 |

| Query_num | Method | pria | aair | ppia | ppip | prip | rpip | rrip | rpia | aain | nain | nair | anin | nnin | nnir | anir | rria |
|---|---|---|---|---|---|---|---|---|---|---|---|---|---|---|---|---|---|
| 77 | GQE+NRN | 19.95 | 0.53 | 16.69 | 14.02 | 12.88 | 11.43 | 7.56 | 14.03 | 0.12 | 0.09 | 0.33 | 0.10 | 0.31 | 0.30 | 0.32 | 8.15 |
| | Q2B+NRN | 17.51 | 0.40 | 19.46 | 11.36 | 14.26 | 14.98 | 11.94 | 10.86 | 0.11 | 0.08 | 0.39 | 0.09 | 0.30 | 0.23 | 0.38 | 6.67 |
| | Q2P+NRN | 35.66 | 0.05 | 18.10 | 5.07 | 8.69 | 8.47 | 7.38 | 20.28 | 0.01 | 0.01 | 0.03 | 0.01 | 0.05 | 0.03 | 0.02 | 15.14 |
| | CNR-NST | 42.49 | 1.72 | 26.49 | 23.29 | 26.71 | 26.85 | 23.82 | 38.54 | 7.69 | 11.20 | 1.53 | 10.72 | 4.76 | 0.88 | 1.38 | 15.14 |
| 92 | GQE+NRN | 14.89 | 0.17 | 16.33 | 13.81 | 12.00 | 12.07 | 8.84 | 14.66 | 4.97 | 6.13 | 0.28 | 6.27 | 0.24 | 0.29 | 0.29 | 3.12 |
| | Q2B+NRN | 14.46 | 0.16 | 17.71 | 12.73 | 11.39 | 12.32 | 10.88 | 18.34 | 5.44 | 5.30 | 0.16 | 5.79 | 0.37 | 0.31 | 0.31 | 2.74 |
| | Q2P+NRN | 20.92 | 0.31 | 15.09 | 19.22 | 17.44 | 17.29 | 16.31 | 16.56 | 4.35 | 3.78 | 0.27 | 3.91 | 0.13 | 0.25 | 0.25 | 1.19 |
| | CNR-NST | 39.19 | 1.58 | 24.12 | 26.16 | 24.42 | 26.62 | 25.56 | 46.20 | 9.70 | 10.82 | 1.57 | 11.57 | 4.54 | 0.77 | 1.37 | 8.90 |

| Query_num | Method | pppi | ppri | arpi | rppi | arri | rpri | nnai | pani | nnni | anni | anai | rani | nrpi | raai | nrri | paai |
|---|---|---|---|---|---|---|---|---|---|---|---|---|---|---|---|---|---|
| 77 | GQE+NRN | 23.16 | 32.30 | 27.97 | 24.52 | - | 28.02 | 40.38 | - | 5.86 | 0.83 | - | 0.23 | 3.03 | 11.56 | - | 18.28 |
| | Q2B+NRN | 19.72 | 35.65 | 33.90 | 26.07 | - | 29.72 | 41.82 | - | 6.12 | 0.85 | - | 0.50 | 2.83 | 20.62 | - | 32.11 |
| | Q2P+NRN | 15.15 | 52.74 | 39.31 | 24.52 | - | 33.18 | 49.27 | - | 12.74 | 0.01 | - | 0.01 | 1.22 | 12.95 | - | 27.96 |
| | CNR-NST | 39.56 | 52.59 | 51.24 | 43.73 | - | 69.50 | 55.74 | - | 12.20 | 5.54 | - | 30.25 | 8.06 | 17.27 | - | 31.27 |
| 92 | GQE+NRN | 22.79 | 31.00 | 35.16 | 23.93 | 46.33 | 34.95 | 43.85 | 30.71 | 10.13 | 0.66 | 7.68 | 38.43 | 3.48 | 12.20 | 42.01 | 19.85 |
| | Q2B+NRN | 19.49 | 40.11 | 28.30 | 23.21 | 42.91 | 32.96 | 43.40 | 30.56 | 8.29 | 0.66 | 6.02 | 36.53 | 2.89 | 20.32 | 39.33 | 33.45 |
| | Q2P+NRN | 32.87 | 38.29 | 38.75 | 32.35 | 36.01 | 31.80 | 40.73 | 19.40 | 9.35 | 0.54 | 5.88 | 27.28 | 1.26 | 27.28 | 36.01 | 30.28 |
| | CNR-NST | 38.01 | 53.34 | 44.27 | 42.35 | 53.94 | 62.77 | 50.33 | 30.62 | 13.82 | 5.04 | 7.14 | 36.67 | 7.45 | 17.01 | 52.27 | 32.65 |

| Query_num | Method | 2pu | rpu | 2ru | pru | 2au | 2nu | anu | nau |
|---|---|---|---|---|---|---|---|---|---|
| 77 | GQE+NRN | 4.79 | 3.70 | 0.40 | 3.70 | 0.39 | 0.31 | 0.16 | 0.16 |
| | Q2B+NRN | 4.27 | 3.82 | 0.34 | 4.33 | 0.49 | 0.71 | 0.53 | 0.58 |
| | Q2P+NRN | 13.76 | 15.35 | 0.04 | 16.03 | 0.03 | 0.01 | 0.05 | 0.01 |
| | CNR-NST | 14.63 | 9.63 | 1.14 | 9.80 | 1.10 | 2.91 | 5.36 | 5.88 |
| 92 | GQE+NRN | 4.50 | 3.39 | 0.39 | 3.46 | 0.41 | 0.39 | 0.16 | 0.18 |
| | Q2B+NRN | 3.31 | 3.69 | 0.24 | 3.99 | 0.37 | 0.57 | 0.31 | 0.33 |
| | Q2P+NRN | 11.63 | 7.07 | 0.23 | 6.07 | 0.14 | 0.25 | 0.45 | 0.42 |
| | CNR-NST | 14.86 | 9.40 | 1.28 | 9.06 | 1.17 | 1.99 | 4.79 | 4.67 |

| Query_num | Method | 2pup | 2pua | prup | prua | anun | anur | 2aur | 2aun | naur | naun | 2nun | 2nur | 2rup | 2rua | rpua | rpup |
|---|---|---|---|---|---|---|---|---|---|---|---|---|---|---|---|---|---|
| 77 | GQE+NRN | 5.90 | 3.75 | 5.71 | 3.19 | 0.25 | 0.24 | 4.24 | 0.11 | 0.27 | 0.25 | 0.46 | 0.42 | 3.79 | 1.68 | 3.49 | 5.75 |
| | Q2B+NRN | 5.07 | 3.11 | 6.03 | 3.57 | 0.37 | 0.21 | 3.60 | 0.13 | 0.22 | 0.39 | 0.95 | 0.34 | 4.26 | 1.64 | 3.54 | 5.68 |
| | Q2P+NRN | 3.00 | 3.36 | 5.36 | 3.13 | 0.01 | 0.06 | 18.16 | 0.01 | 0.08 | 0.01 | 0.01 | 0.05 | 7.57 | 0.60 | 3.62 | 4.88 |
| | CNR-NST | 16.57 | 10.43 | 14.50 | 9.21 | 2.03 | 0.49 | 1.35 | 4.45 | 0.54 | 2.60 | 1.58 | 0.53 | 11.15 | 2.26 | 8.28 | 13.25 |
| 92 | GQE+NRN | 7.70 | 3.68 | 6.06 | 4.11 | 0.71 | 0.24 | 4.16 | 0.45 | 0.27 | 0.69 | 0.40 | 0.42 | 3.32 | 1.90 | 3.27 | 5.11 |
| | Q2B+NRN | 6.27 | 2.69 | 6.23 | 2.94 | 1.11 | 0.15 | 2.12 | 0.40 | 0.18 | 0.91 | 0.87 | 0.21 | 3.50 | 1.48 | 3.14 | 5.28 |
| | Q2P+NRN | 9.82 | 2.72 | 8.84 | 1.92 | 0.58 | 0.20 | 5.08 | 0.24 | 0.23 | 0.56 | 0.45 | 0.34 | 5.84 | 1.92 | 2.15 | 7.66 |
| | CNR-NST | 18.20 | 10.56 | 15.05 | 7.89 | 2.54 | 0.44 | 1.38 | 3.98 | 0.33 | 2.39 | 1.47 | 0.33 | 12.44 | 2.53 | 8.83 | 14.86 |

| Query_num | Method | 2b | 3b | bp | | pb | | 2pb | | | |
|---|---|---|---|---|---|---|---|---|---|---|---|
| | | aab | aaab | aabr | aabn | raab | paab | rarab | rapab | parab | papab |
| 102 | CNR-NST | 11.71 | 12.49 | 0.85 | 11.00 | 13.49 | 24.40 | 13.29 | 21.67 | 25.85 | 22.33 |

## D.2 DETAILED EXPERIMENTAL RESULTS OF THE NEW EVALUATION METRICS

Table 15: Detailed MRR_0.001 Results (%) in all Test Set

| Dataset | Method | 1ap | 1np | an | pa | ra | 2n | 2ai | ani | 2ni | nai |
|---|---|---|---|---|---|---|---|---|---|---|---|
| FB15K | Q2P+NRN | 0.79 | 0.11 | 0.12 | 0.97 | 1.07 | 0.11 | 1.11 | 2.02 | 0.31 | 2.31 |
| | CNR-NST | 20.56 | 24.51 | 25.70 | 15.66 | 8.96 | 15.30 | 66.03 | 26.65 | 27.15 | 25.21 |
| DB15K | Q2P+NRN | 8.87 | 1.06 | 2.83 | 7.83 | 12.01 | 2.48 | 8.87 | 2.96 | 3.58 | 2.89 |
| | CNR-NST | 17.41 | 20.42 | 17.19 | 17.65 | 17.77 | 9.19 | 46.09 | 25.03 | 17.95 | 13.42 |
| YAGO15K | Q2P+NRN | 4.53 | 6.78 | 20.51 | 5.93 | 2.81 | 55.10 | 5.32 | 4.81 | 21.52 | 6.26 |
| | CNR-NST | 19.06 | 17.64 | 21.23 | 19.01 | 12.12 | 17.20 | 45.76 | 31.11 | 20.68 | 34.12 |

| Dataset | Method | paai | pani | anni | anai | nnai | nnni | rani | raai |
|---|---|---|---|---|---|---|---|---|---|
| FB15K | Q2P+NRN | 1.72 | 2.02 | 0.45 | 1.32 | 1.37 | 0.40 | 0.91 | 1.53 |
| | CNR-NST | 63.62 | 25.76 | 27.27 | 46.29 | 39.46 | 18.67 | 19.96 | 61.60 |
| DB15K | Q2P+NRN | 7.74 | 6.78 | 3.00 | 3.28 | 3.14 | 5.30 | 20.35 | 27.71 |
| | CNR-NST | 56.27 | 21.29 | 26.04 | 44.89 | 51.48 | 14.22 | 29.59 | 43.30 |
| YAGO15K | Q2P+NRN | 6.95 | 8.26 | 24.14 | 5.22 | 4.88 | 45.47 | 4.43 | 5.13 |
| | CNR-NST | 55.52 | 20.46 | 22.99 | 38.58 | 38.17 | 21.62 | 17.95 | 57.47 |

| Dataset | Method | 2pia | pria | 2ain | anin | rpia | 2ria | 2nin | nain |
|---|---|---|---|---|---|---|---|---|---|
| FB15K | Q2P+NRN | 1.42 | 0.96 | 0.11 | 0.13 | 0.72 | 0.03 | 0.12 | 0.13 |
| | CNR-NST | 20.63 | 35.04 | 22.28 | 28.54 | 32.89 | 6.42 | 12.02 | 23.07 |
| DB15K | Q2P+NRN | 6.25 | 16.97 | 7.51 | 3.25 | 20.18 | 44.58 | 6.49 | 4.30 |
| | CNR-NST | 23.59 | 34.34 | 7.39 | 16.13 | 40.18 | 56.93 | 8.53 | 12.08 |
| YAGO15K | Q2P+NRN | 6.20 | 2.91 | 27.87 | 39.73 | 1.98 | 0.81 | 56.38 | 38.59 |
| | CNR-NST | 23.67 | 22.41 | 15.44 | 26.53 | 26.21 | 22.40 | 17.42 | 24.89 |

| Dataset | Method | 2na3i | 3n3i | nan3i | n2a3i | 2an3i | 3a3i | a2n3i | ana3i |
|---|---|---|---|---|---|---|---|---|---|
| FB15K | Q2P+NRN | 2.85 | 0.50 | 3.18 | 2.05 | 2.86 | 11.27 | 3.11 | 2.37 |
| | CNR-NST | 27.19 | 32.68 | 36.71 | 53.68 | 57.51 | 78.05 | 35.18 | 54.86 |
| DB15K | Q2P+NRN | 2.14 | 7.04 | 1.99 | 8.00 | 6.37 | 7.92 | 3.59 | 7.72 |
| | CNR-NST | 19.93 | 23.69 | 20.74 | 35.90 | 32.97 | 54.71 | 25.70 | 39.22 |
| YAGO15K | Q2P+NRN | 7.42 | 36.03 | 7.73 | 7.22 | 6.53 | 5.23 | 7.68 | 5.50 |
| | CNR-NST | 31.78 | 26.41 | 28.08 | 47.05 | 45.61 | 55.37 | 28.83 | 37.60 |

| Dataset | Method | aau | 2nu | nau | 2pua | prua | anun | 2aun | naun | 2nun | 2rua | rpua |
|---|---|---|---|---|---|---|---|---|---|---|---|---|
| FB15K | Q2P+NRN | 0.47 | 0.29 | 0.26 | 1.93 | 1.66 | 0.28 | 0.28 | 0.36 | 0.34 | 0.74 | 1.64 |
| | CNR-NST | 4.91 | 6.22 | 11.86 | 11.29 | 14.05 | 5.25 | 10.32 | 6.30 | 3.15 | 11.44 | 12.25 |
| DB15K | Q2P+NRN | 3.76 | 27.26 | 23.82 | 10.92 | 22.80 | 23.24 | 27.93 | 22.11 | 21.73 | 26.77 | 22.64 |
| | CNR-NST | 6.51 | 20.87 | 24.99 | 16.80 | 19.57 | 20.23 | 29.34 | 24.94 | 17.87 | 26.63 | 26.22 |
| YAGO15K | Q2P+NRN | 2.96 | 57.48 | 56.71 | 6.34 | 5.57 | 60.38 | 62.93 | 59.76 | 63.90 | 3.08 | 6.18 |
| | CNR-NST | 9.37 | 26.67 | 33.67 | 13.43 | 13.25 | 20.12 | 33.21 | 20.13 | 14.54 | 8.79 | 11.80 |

## E EXAMPLES OF NUMERIC QUERIES

This section presents a subset of query types from the 102 sub-queries that involve numerical reasoning and computation. We selected several queries with real-world significance to demonstrate that most of our query types are capable of reflecting real-world scenarios.

For some query answers (e.g., 1ap), there is only one correct answer, but we still present the top 5 inferred answers. Additionally, for queries with too many answers, we skipped some of the correctly predicted easy answers to focus on demonstrating that our model can still infer the correct hard answers.

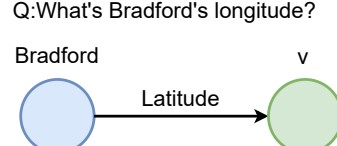

Q:What's Bradford's longitude?

| Logical Expression: | $Q[N_?] = N_?, \exists N_?, a_1(e_1, N_?)$ | | |
|---|---|---|---|
| Rank | Query Answers | Correctness | Answer type |
| 1 | 53.69 | ✗ | - |
| 2 | 53.8 | ✔ | Hard |
| 3 | 51.5 | ✗ | - |
| 4 | 51.48 | ✗ | - |
| 5 | 51.51 | ✗ | - |

Figure 8: Intermediate variable assignments and ranks for example 1ap query.

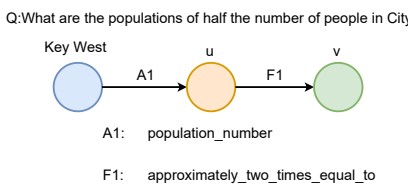

Q:What are the years earlier than 2003?

| Logical Expression: | $Q[N_?] = N_?, \exists N_?, f_1(n_1, N_?)$ | | |
|---|---|---|---|
| Rank | Query Answers | Correctness | Answer type |
| 1 | 2002.17 | ✔ | Easy |
| 2 | 1988.33 | ✔ | Easy |
| 3 | 72202 | ✗ | - |
| 4 | 2003 | ✗ | - |
| 5 | 1920.92 | ✗ | - |
| 6 | 1963.92 | ✔ | Easy |
| 7 | 10934.93 | ✗ | - |
| 8 | 1911.5 | ✗ | - |
| 9 | 1950.83 | ✔ | Easy |
| 10 | 1977.33 | ✔ | Easy |
| 11 | 1948.67 | ✔ | Easy |
| 12 | 1974.5 | ✔ | Easy |
| 13 | 1948.92 | ✔ | Hard |

Figure 9: Intermediate variable assignments and ranks for example 1np query.

Q:What are the populations of half the number of people in City A?

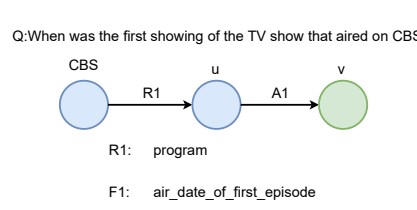

A1: population_number

F1: approximately_two_times_equal_to

| Logical Expression: | $Q[N_?] = N_?, \exists N_a, a_1(e_1, N_a) \land f_1(N_a, N_?)$ | | |
|---|---|---|---|
| Rank | Query Answers | Correctness | Answer type |
| 1 | 53827.00 | ✔ | Easy |
| 2 | 52966.00 | ✔ | Easy |
| 3 | 53326.00 | ✔ | Easy |
| 4 | 53066.00 | ✔ | Easy |
| 5 | 53025.00 | ✔ | Hard |
| 6 | 53374.00 | ✔ | Easy |
| 7 | 53437.00 | ✔ | Easy |
| 8 | 53483.00 | ✗ | - |
| 9 | 53838.00 | ✔ | Easy |
| 10 | 53311.00 | ✔ | Easy |

Figure 10: Intermediate variable assignments and ranks for example an query.

Q:When was the first showing of the TV show that aired on CBS?

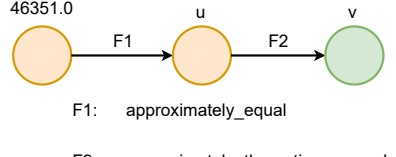

R1: program

F1: air_date_of_first_episode

| Logical Expression: | $Q[N_?] = N_?, \exists E_a, r_1(e_1, E_a) \land a_1(E_a, N_?)$ | | |
|---|---|---|---|
| Rank | Query Answers | Correctness | Answer type |
| 1 | 1978.32 | ✔ | Easy |
| 2 | 1992.75 | ✔ | Easy |
| 3 | 1985.75 | ✔ | Easy |
| 4 | 1993.67 | ✔ | Easy |
| 5 | 1990.75 | ✔ | Easy |
| ..... | ..... | ..... | ..... |
| 26 | 1960.83 | ✔ | Hard |
| 27 | 1996.75 | ✔ | Hard |
| 28 | 2000.42 | ✗ | - |
| 29 | 1959.83 | ✔ | Easy |
| 30 | 1972.75 | ✔ | Easy |

Figure 11: Intermediate variable assignments and ranks for example pa query.

F1: approximately_equal

F2: approximately_three_times_equal_to

| Logical Expression: | $Q[N_?] = N_?, \exists N_a, f_1(n_1, N_a) \land f_2(N_a, N_?)$ | | |
|---|---|---|---|
| Rank | Query Answers | Correctness | Answer type |
| 1 | 137776 | ✔ | Hard |
| 2 | 138296 | ✔ | Easy |
| 3 | 137555 | ✔ | Easy |
| 4 | 139790 | ✔ | Easy |
| 5 | 139390 | ✗ | - |

Figure 12: Intermediate variable assignments and ranks for example 2n query.

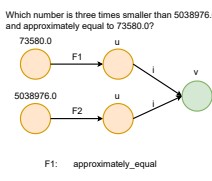

| Logical Expression: | $Q[N_?] = N_?, \exists N_?, f_1(n_1, N_?) \wedge f_2(n_2, N_?)$ | | |
|---|---|---|---|
| Rank | Query Answers | Correctness | Answer type |
| 1 | 73615.00 | ✔ | Hard |
| 2 | 73485.00 | ✔ | Easy |
| 3 | 73208.00 | ✔ | Easy |
| 4 | -77.91 | ✗ | - |
| 5 | -85.17 | ✗ | - |

Figure 13: Intermediate variable assignments and ranks for example 2ni query.

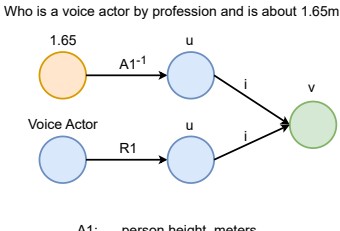

| Logical Expression: | $Q[E_?] = E_?, \exists E_?, a_1^{-1}(n_1, E_?) \wedge r_1(e_1, E_?)$ | | |
|---|---|---|---|
| Rank | Query Answers | Correctness | Answer type |
| 1 | Debi Mazar | ✔ | Easy |
| 2 | Andrea Bowen | ✔ | Easy |
| 3 | Ian Holm | ✔ | Easy |
| 4 | Breckin Meyer | ✔ | Easy |
| 5 | Mel Brooks | ✔ | Easy |
| 6 | Christina Applegate | ✔ | Easy |
| 7 | Roy Kinnear | ✔ | Easy |
| 8 | Nathan Lane | ✔ | Hard |
| 9 | Molly Shannon | ✗ | - |
| 10 | Common | ✗ | - |
| 11 | Anna Paquin | ✔ | Hard |

Figure 14: Intermediate variable assignments and ranks for example rpi query.

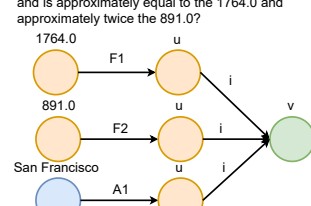

| Logical Expression: | $Q[N_?] = N_?, \exists N_?, a_1(e_1, N_?) \wedge f_1(n_1, N_?) \wedge f_2(n_2, N_?)$ | | |
|---|---|---|---|
| Rank | Query Answers | Correctness | Answer type |
| 1 | 1776.50 | ✔ | Hard |
| 2 | 1763.83 | ✗ | - |
| 3 | 1765.00 | ✗ | - |
| 4 | 1775.67 | ✗ | - |
| 5 | 1776.67 | ✗ | - |

Figure 15: Intermediate variable assignments and ranks for example 2na3i query.

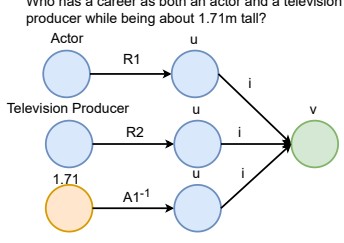

| Logical Expression: | $Q[E_?] = E_?, \exists E_?, r_1(e_1, E_?) \wedge r_2(e_2, E_?) \wedge a_1^{-1}(n_1, E_?)$ | | |
|---|---|---|---|
| Rank | Query Answers | Correctness | Answer type |
| 1 | Ellen DeGeneres | ✔ | Easy |
| 2 | Candice Bergen | ✔ | Easy |
| 3 | Joan Chen | ✔ | Easy |
| 4 | Martin Lawrence | ✔ | Easy |
| 5 | Martin Short | ✔ | Easy |
| 6 | Kirstie Alley | ✔ | Hard |

Figure 16: Intermediate variable assignments and ranks for example 2pr3i query.

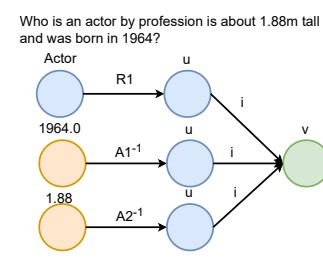

Who is an actor by profession is about 1.88m tall and was born in 1964?

R1: people_with_this_profession

A1: person.date_of_birth

A2: person.height_meters

| Logical Expression: | $Q\left[E_{?}\right]=E_{?},\exists E_{?},r_{1}\left(e_{1},E_{?}\right)\wedge a_{1}^{-1}\left(n_{1},E_{?}\right)\wedge a_{2}^{-1}\left(n_{2},E_{?}\right)$ | | |
|---|---|---|---|
| Rank | Query Answers | Correctness | Answer type |
| 1 | Brendan Coyle | ✔ | Easy |
| 2 | Benjamin Bratt | ✔ | Hard |
| 3 | Til Schweiger | ✗ | - |
| 4 | Iqbal Theba | ✗ | - |
| 5 | Ray Romano | ✗ | - |

Figure 17: Intermediate variable assignments and ranks for example p2r3i query.

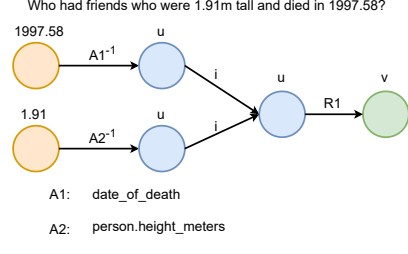

Who had friends who were 1.91m tall and died in 1997.58?

A1: date_of_death

A2: person.height_meters

R1: friends

| Logical Expression: | $Q\left[E_{?}\right]=E_{?},\exists E_{?},r_{1}\left(E_{1},E_{?}\right)\wedge\left(\exists E_{1},a_{1}^{-1}\left(n_{1},E_{1}\right)\wedge a_{2}^{-1}\left(n_{2},E_{1}\right)\right)$ | | |
|---|---|---|---|
| Rank | Query Answers | Correctness | Answer type |
| 1 | Henry Fonda | ✔ | Easy |
| 2 | Robert Taylor | ✔ | Easy |
| 3 | Myrna Loy | ✔ | Easy |
| 4 | Clark Gable | ✔ | Easy |
| 5 | Robert Young | ✔ | Easy |
| 6 | Joan Crawford | ✔ | Easy |
| 7 | Loretta Young | ✔ | Easy |
| 8 | James Stewart | ✗ | - |
| 9 | Bette Davis | ✔ | Hard |
| 10 | Lucille Ball | ✗ | - |

Figure 18: Intermediate variable assignments and ranks for example 2rip query.

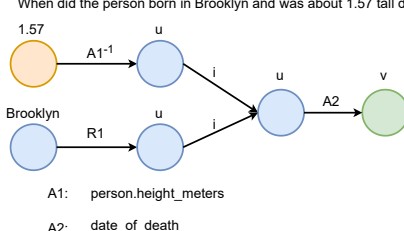

When did the person born in Brooklyn and was about 1.57 tall die?

A1: person.height_meters

A2: date_of_death

R1: people_born_here

| Logical Expression: | $Q\left[N_{?}\right]=N_{?},\exists N_{?},a_{1}\left(E_{1},N_{?}\right)\wedge\left(\exists E_{1},a_{2}^{-1}\left(n_{1},E_{1}\right)\wedge r_{1}\left(e_{1},E_{1}\right)\right)$ | | |
|---|---|---|---|
| Rank | Query Answers | Correctness | Answer type |
| 1 | 2014.75 | ✔ | Easy |
| 2 | 2014.33 | ✔ | Hard |
| 3 | 2014.17 | ✗ | - |
| 4 | 2001.67 | ✗ | - |
| 5 | 2011.42 | ✗ | - |

Figure 19: Intermediate variable assignments and ranks for example rpia query.

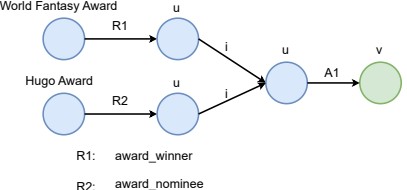

When was the death of the person who won both the World Fantasy Award and was nominated for the Hugo Award?

R1: award_winner

R2: award_nominee

A1: date_of_birth

| Logical Expression: | $Q\left[N_{?}\right]=N_{?},\exists N_{?},a_{1}\left(E_{1},N_{?}\right)\wedge\left(\exists E_{1},r_{1}\left(e_{1},E_{1}\right)\wedge r_{2}\left(e_{2},E_{1}\right)\right)$ | | |
|---|---|---|---|
| Rank | Query Answers | Correctness | Answer type |
| 1 | 1916.67 | ✔ | Easy |
| 2 | 1929.83 | ✔ | Easy |
| 3 | 1934.43 | ✔ | Hard |
| 4 | 1911 | ✔ | Hard |
| 5 | 1947.75 | ✗ | - |

Figure 20: Intermediate variable assignments and ranks for example 2pia query.

What is the race of the people in the film Eragon whose actors are about 1.7m tall?

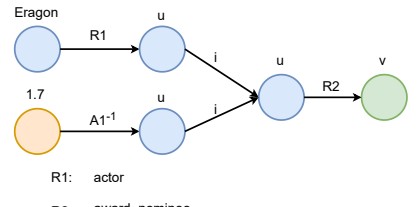

| Logical Expression: | $Q\,[E_?] = E_?, \exists E_?, r_1\,(E_1, E_?) \wedge (\exists E_1, a_1^{-1}\,(n_1, E_1) \wedge r_2\,(e_2, E_1))$ | | |
|---|---|---|---|
| Rank | Query Answers | Correctness | Answer type |
| 1 | English people | ✔ | Easy |
| 2 | Jewish people | ✔ | Easy |
| 3 | Ashkenazi Jews | ✔ | Easy |
| 4 | Hungarians | ✔ | Hard |
| 5 | White Americans | ✗ | - |

R1: actor

R2: award_nominee

A1: person_ethnicity

Figure 21: Intermediate variable assignments and ranks for example rpip query.

Which films are based on Shawn Wayans and were released around 2003?

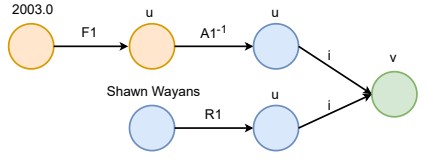

| Logical Expression: | $Q\,[E_?] = E_?, \exists E_?, r_1\,(E_1, E_?) \wedge (\exists E_?, f_1\,(n_1, N_1) \wedge a_1^{-1}\,(N_1, E_?))$ | | |
|---|---|---|---|
| Rank | Query Answers | Correctness | Answer type |
| 1 | Scary Movie 3 | ✔ | Easy |
| 2 | Pride & Prejudice | ✔ | Easy |
| 3 | Scary Movie 4 | ✔ | Hard |
| 4 | Scary Movie | ✗ | - |
| 5 | Kill Bill Volume 1 | ✗ | - |

F1: approximately_equal

R1: film_story_contributor

A1: film.initial_release_date

Figure 22: Intermediate variable assignments and ranks for example nrpi query.

Who have won awards founded in 1988 and are 1.88m tall?

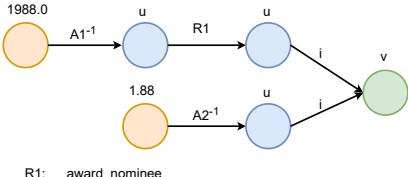

| Logical Expression: | $Q\,[E_?] = E_?, \exists E_?, a_1^{-1}\,(n_1, E_?) \wedge (\exists E_?, a_2^{-1}\,(n_2, E_1) \wedge r_1\,(E_1, E_?))$ | | |
|---|---|---|---|
| Rank | Query Answers | Correctness | Answer type |
| 1 | James Coburn | ✔ | Easy |
| 2 | Ray McKinnon | ✔ | Easy |
| 3 | Djimon Hounsou | ✔ | Hard |
| 4 | Morgan Freeman | ✔ | Hard |
| 5 | Danny Huston | ✗ | - |

R1: award_nominee

A1: location.date_founded

A2: person.height_meters

Figure 23: Intermediate variable assignments and ranks for example rpri query.

Which NFL teams have the all-time superstar position of punter and have players born in 1962.83?

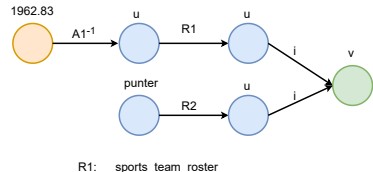

| Logical Expression: | $Q\,[E_?] = E_?, \exists E_?, r_1\,(e_1, E_?) \wedge (\exists E_?, a_2^{-1}\,(n_2, E_1) \wedge r_2\,(E_1, E_?))$ | | |
|---|---|---|---|
| Rank | Query Answers | Correctness | Answer type |
| 1 | New England Patriots | ✔ | Easy |
| 2 | Buffalo Bills | ✔ | Easy |
| 3 | Los Angeles Chargers | ✔ | Easy |
| 4 | Chicago Bears | ✔ | Hard |
| 5 | Toronto Argonauts | ✗ | - |

R1: sports_team_roster

R2: football_historical_roster_position

A1: person.date_of_birth

Figure 24: Intermediate variable assignments and ranks for example rppi query.

Which films are in the same genre as Sideways and were released in 2008.75?

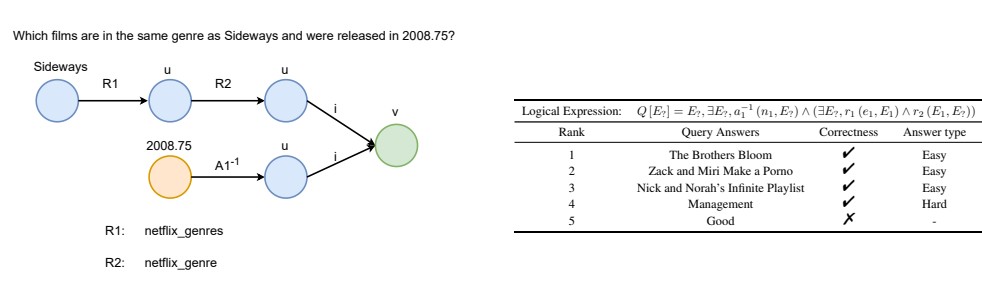

| Logical Expression: | $Q[E_?] = E_?, \exists E_?, a_1^{-1}(n_1, E_?) \wedge (\exists E_?, r_1(e_1, E_1) \wedge r_2(E_1, E_?))$ | | |
|---|---|---|---|
| Rank | Query Answers | Correctness | Answer type |
| 1 | The Brothers Bloom | ✔ | Easy |
| 2 | Zack and Miri Make a Porno | ✔ | Easy |
| 3 | Nick and Norah's Infinite Playlist | ✔ | Easy |
| 4 | Management | ✔ | Hard |
| 5 | Good | ✗ | - |

R1: netflix_genres

R2: netflix_genre

A1: film.initial_release_date

Figure 25: Intermediate variable assignments and ranks for example ppri query.

Which number is the release date of Breakfast at Tiffany's or approximately equal to 23398.0?

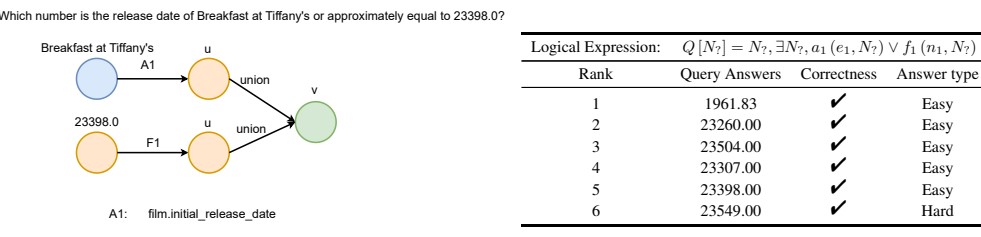

| Logical Expression: | $Q[N_?] = N_?, \exists N_?, a_1(e_1, N_?) \vee f_1(n_1, N_?)$ | | |
|---|---|---|---|
| Rank | Query Answers | Correctness | Answer type |
| 1 | 1961.83 | ✔ | Easy |
| 2 | 23260.00 | ✔ | Easy |
| 3 | 23504.00 | ✔ | Easy |
| 4 | 23307.00 | ✔ | Easy |
| 5 | 23398.00 | ✔ | Easy |
| 6 | 23549.00 | ✔ | Hard |

A1: film.initial_release_date

F1: three_times_larger_than

Figure 26: Intermediate variable assignments and ranks for example anu query.

Which entities are the subject of Austin Powers: International Man of Mystery or cities with a longitude of -111.94 degrees?

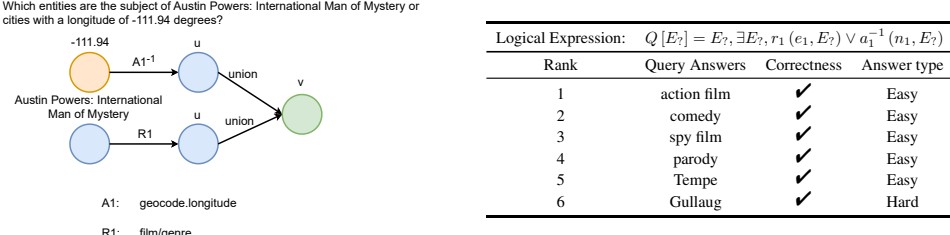

| Logical Expression: | $Q[E_?] = E_?, \exists E_?, r_1(e_1, E_?) \vee a_1^{-1}(n_1, E_?)$ | | |
|---|---|---|---|
| Rank | Query Answers | Correctness | Answer type |
| 1 | action film | ✔ | Easy |
| 2 | comedy | ✔ | Easy |
| 3 | spy film | ✔ | Easy |
| 4 | parody | ✔ | Easy |
| 5 | Tempe | ✔ | Easy |
| 6 | Gullaug | ✔ | Hard |

A1: geocode.longitude

R1: film/genre

Figure 27: Intermediate variable assignments and ranks for example rpu query.

Which events were organised in 1927.33 or 2012.17?

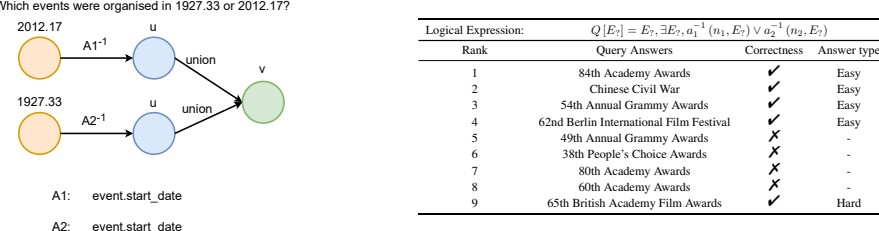

| Logical Expression: | $Q[E_?] = E_?, \exists E_?, a_1^{-1}(n_1, E_?) \vee a_2^{-1}(n_2, E_?)$ | | |
|---|---|---|---|
| Rank | Query Answers | Correctness | Answer type |
| 1 | 84th Academy Awards | ✔ | Easy |
| 2 | Chinese Civil War | ✔ | Easy |
| 3 | 54th Annual Grammy Awards | ✔ | Easy |
| 4 | 62nd Berlin International Film Festival | ✔ | Easy |
| 5 | 49th Annual Grammy Awards | ✗ | - |
| 6 | 38th People's Choice Awards | ✗ | - |
| 7 | 80th Academy Awards | ✗ | - |
| 8 | 60th Academy Awards | ✗ | - |
| 9 | 65th British Academy Film Awards | ✔ | Hard |

A1: event.start_date

A2: event.start_date

Figure 28: Intermediate variable assignments and ranks for example 2ru query.

What is the longitude of the area in which Carol Burnett's institution or the film in which Martin Freeman appeared was released?

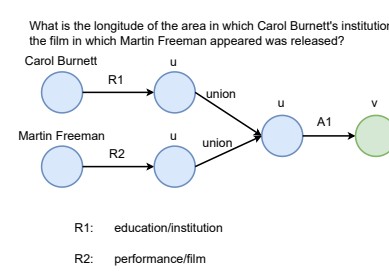

R1: education/institution

R2: performance/film

A1: geocode.longitude

| Logical Expression: | $Q[N_?] = N_?, \exists N_?, a_1(E_1, N_?) \wedge (\exists E_1, r_1(e_1, E_1) \vee r_2(e_2, E_1))$ | | |
|---|---|---|---|
| Rank | Query Answers | Correctness | Answer type |
| 1 | -118.34 | ✔ | Easy |
| 2 | -118.44 | ✔ | Hard |
| 3 | -97.74 | ✗ | - |
| 4 | -87.67 | ✗ | - |
| 5 | -72.93 | ✗ | - |

Figure 29: Intermediate variable assignments and ranks for example 2pua query.

What musical compositions or companies came from places neighbouring Ashtabula County or from people who died in 1941.08?

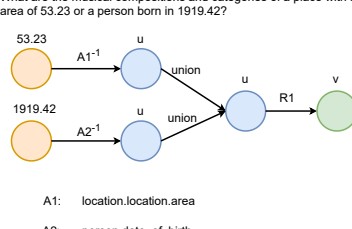

R1: adjoins

R2: music/artist/label

A1: person.date_of_death

| Logical Expression: | $Q[E_?] = E_?, \exists E_?, r_1(E_1, E_?) \wedge (\exists E_1, r_1(e_1, E_1) \vee a_1^{-1}(n_1, E_1))$ | | |
|---|---|---|---|
| Rank | Query Answers | Correctness | Answer type |
| 1 | Virgin Records | ✔ | Easy |
| 2 | Columbia Records | ✔ | Easy |
| 3 | Vanguard Records | ✔ | Easy |
| 4 | Koch Entertainment | ✔ | Easy |
| 5 | Warner Bros. Records | ✗ | - |
| 6 | MCA Records | ✔ | Hard |

Figure 30: Intermediate variable assignments and ranks for example rpup query.

What are the musical compositions and categories of a place with an area of 53.23 or a person born in 1919.42?

A1: location.location.area

A2: person.date_of_birth

R1: /music/artist/label

| Logical Expression: | $Q[E_?] = E_?, \exists E_?, r_1(E_1, E_?) \wedge (\exists E_1, a_1^{-1}(n_1, E_1) \vee a_2^{-1}(n_2, E_1))$ | | |
|---|---|---|---|
| Rank | Query Answers | Correctness | Answer type |
| 1 | Columbia Records | ✔ | Easy |
| 2 | Vanguard Records | ✔ | Hard |
| 3 | Warner Bros. Records | ✗ | - |
| 4 | EMI | ✗ | - |

Figure 31: Intermediate variable assignments and ranks for example 2rup query.

What is the height of a person born in 1964.33 or a person born in 1951.58?

A1: person.date_of_birth

A2: person.date_of_birth

A3: person.height_meters

| Logical Expression: | $Q[N_?] = N_?, \exists N_?, a_1(E_1, N_?) \wedge (\exists E_1, a_2^{-1}(n_1, E_1) \vee a_3^{-1}(n_2, E_1))$ | | |
|---|---|---|---|
| Rank | Query Answers | Correctness | Answer type |
| 1 | 1.75 | ✔ | Easy |
| 2 | 1.70 | ✔ | Easy |
| 3 | 1.88 | ✔ | Easy |
| 4 | 1.68 | ✔ | Easy |
| 5 | 1.85 | ✔ | Easy |
| 6 | 1.78 | ✔ | Easy |
| 7 | 1.82 | ✔ | Easy |
| 8 | 1.73 | ✗ | - |
| 9 | 1.65 | ✔ | Hard |

Figure 32: Intermediate variable assignments and ranks for example 2rua query.

What is the longitude of a film or television production released in 1912.0 or the capital of Durham?

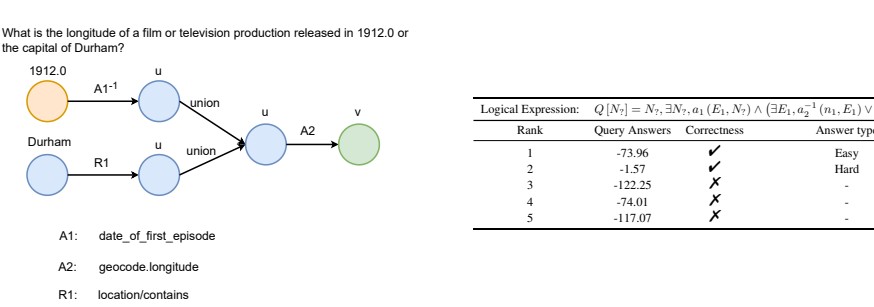

| Logical Expression: | $Q[N_?] = N_?, \exists N_?, a_1(E_1, N_?) \wedge (\exists E_1, a_2^{-1}(n_1, E_1) \vee r_1(e_1, E_1))$ | | |
|---|---|---|---|
| Rank | Query Answers | Correctness | Answer type |
| 1 | -73.96 | ✔ | Easy |
| 2 | -1.57 | ✔ | Hard |
| 3 | -122.25 | ✗ | - |
| 4 | -74.01 | ✗ | - |
| 5 | -117.07 | ✗ | - |

A1: date_of_first_episode

A2: geocode.longitude

R1: location/contains

Figure 33: Intermediate variable assignments and ranks for example rpua query.

What is the sum of Lubbock's total population and Hamilton's founding date?

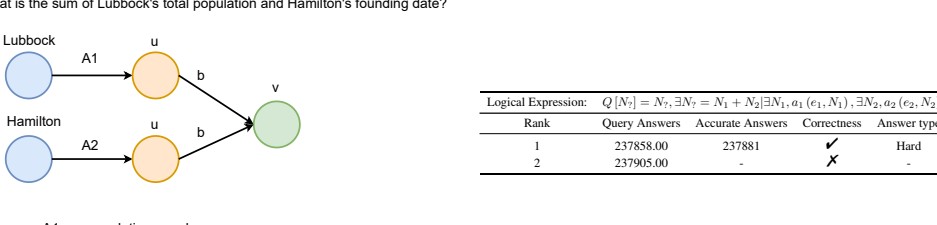

| Logical Expression: | $Q[N_?] = N_?, \exists N_? = N_1 + N_2 | \exists N_1, a_1(e_1, N_1), \exists N_2, a_2(e_2, N_2)$ | | |
|---|---|---|---|
| Rank | Query Answers | Accurate Answers | Correctness | Answer type |
| 1 | 237858.00 | 237881 | ✔ | Hard |
| 2 | 237905.00 | - | ✗ | - |

A1: population_number

A2: dated_location.date_founded

Figure 34: Intermediate variable assignments and ranks for example aab query.

What is the number that approximately equals the total population of Syracuse to the longitude of the Inner Hebrides?

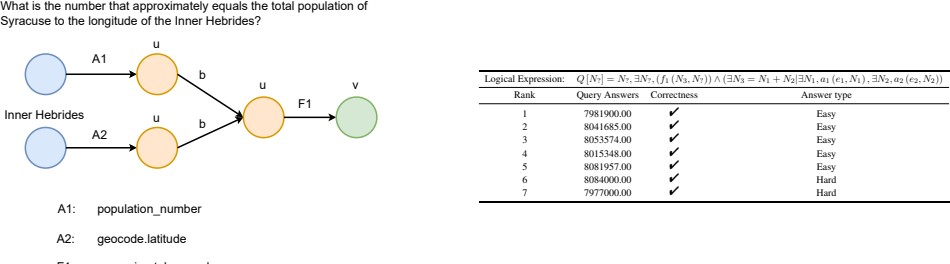

| Logical Expression: | $Q[N_?] = N_?, \exists N_?, (f_1(N_3, N_?)) \wedge (\exists N_3 = N_1 + N_2 | \exists N_1, a_1(e_1, N_1), \exists N_2, a_2(e_2, N_2))$ | |
|---|---|---|
| Rank | Query Answers | Correctness | Answer type |
| 1 | 7981900.00 | ✔ | Easy |
| 2 | 8041685.00 | ✔ | Easy |
| 3 | 8053574.00 | ✔ | Easy |
| 4 | 8015348.00 | ✔ | Easy |
| 5 | 8081957.00 | ✔ | Easy |
| 6 | 8084000.00 | ✔ | Hard |
| 7 | 7977000.00 | ✔ | Hard |

A1: population_number

A2: geocode.latitude

F1: approximately_equal

Figure 35: Intermediate variable assignments and ranks for example aabn query.

What is the sum of the date of birth of the person who won an Olympic medal with A and the longitude of Rockville?

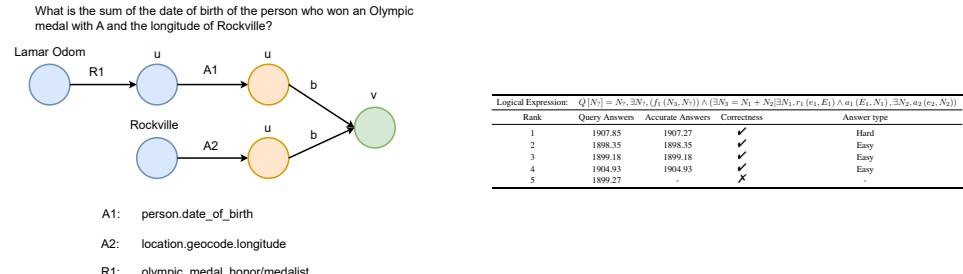

| Logical Expression: | $Q[N_?] = N_?, \exists N_?, (f_1(N_3, N_?)) \wedge (\exists N_3 = N_1 + N_2 | \exists N_1, r_1(e_1, E_1) \wedge a_1(E_1, N_1), \exists N_2, a_2(e_2, N_2))$ | | |
|---|---|---|---|
| Rank | Query Answers | Accurate Answers | Correctness | Answer type |
| 1 | 1907.85 | 1907.27 | ✔ | Hard |
| 2 | 1898.35 | 1898.35 | ✔ | Easy |
| 3 | 1899.18 | 1899.18 | ✔ | Easy |
| 4 | 1904.93 | 1904.93 | ✔ | Easy |
| 5 | 1899.27 | - | ✗ | - |

A1: person.date_of_birth

A2: location.geocode.longitude

R1: olympic_medal_honor/medalist

Figure 36: Intermediate variable assignments and ranks for example paab query.

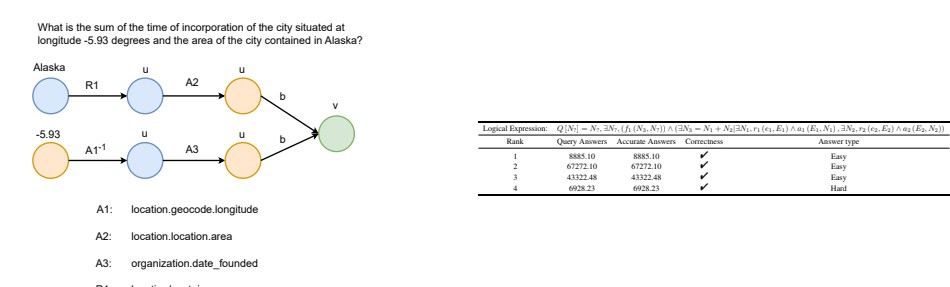

Figure 37: Intermediate variable assignments and ranks for example parab query.

