# OpenReview forum: "Complex Numerical Computation  with Numerical Semantic Pre-training Framework"
_ICLR.cc/2025/Conference — Submitted to ICLR 2025_

### Official Review · Reviewer_KwpT · 2024-10-24

**Soundness:** 2
**Presentation:** 1
**Contribution:** 3
**Rating:** 3
**Confidence:** 4

**Summary:**

The paper aims to solve the task of multi-hop queries over knowledge graphs containing both entities and numerical values. They adapted ComplEx to devise a series of encoders which are trained and then brought together to form a system that, using fuzzy logic, can answer queries on such KGs (including queries that have answers from the real numbers). They evaluate the model across 3 benchmarks datasets against an existing method and show a significant improvement on previous results.

**Strengths:**

1. The improvements on the baseline are very significant, with strong results reported across 3 datasets and a large variety of query types.

2. The ablation study demonstrates well how different parts of their proposed architecture contribute towards its success.

3. The related work is thorough.

**Weaknesses:**

Major:
1. The soundness of the mathematical presentation is very poor: this is my most significant issue with the paper. Many terms and symbols are used without being defined, notation switches arbitrarily, some sections do not make logical sense, and symbols do not match up with standard mathematical practice. As such, it was impossible for me to ascertain precisely how the proposed model operates. More specific details on this can be found below.

2. The paper claims to be evaluating against 3 different numerical reasoning models, but they all come from the same paper and are variants of one another. As a result, I find the evaluation to be lacking, and suggest that the authors evaluate against some other baselines, such as the ones mentioned in L508 - L515, and L524 - L527. Furthermore, the comparison on L494 between training and testing times is not relevant, since evaluation speed is most pertinent when a model is being applied.

3. Sections of the paper are full of clear typos, making it difficult to read. Some of the formatting and placement of various bits of information could also use some restructuring. More details in the minor comments.

Specific concerns with the mathematics:
1. L150 If epsilon is entities, then what is V? And if R is all relations, where are the facts specified in the KG?
2. L154 notation for N mismatches with the one used earlier, has not been properly defined, and usually picks out the natural numbers
3. L172 - L178 "bf" is bad notation, since two variables are being used for one concept.
4. L177 the definition of bf does not parse. Furthermore, it was already defined on L172, so has been defined in two different ways
5. L172 N is not defined
6. L220 - L221 what do all of these arguments refer to?
7. L226 beta is not defind
8. L232 R', A', F' are not defined
9. L238 which normalisation function?
10. L251 (())
11. L249 - L255 I don't see how this defines the above matrices
12. L259 which matrix M?
13. L272 - L287 this section is very confusing, and needs more clarity as to what it is actually describing
14. L295 what is |x| here?
15. L302 what are Vim, u_m and n_m?
16. L309 - L310 wrong notation for cross product and real numbers. And what is F?

Also, what is the signature of the KG?

Other minor concerns:
1. L38 could use a citation
2. L41 "query"
3. L58 "are"
4. L61 "attribute"
5. L108 would be nice to try cut some content to bring this line onto the previous page
6. L140 - L145 is not a contribution, bur rather a result, and does not belong in this section
7. L150, L151, L157, L159, L166, L238, L259, L278 - no space after full stop or comma
8. L153 extract bracket
9. L156 - L160 list not formatted properly, and can be defined more concisely
10. L186 "scored"
11. L188 "t- connorm"
12. L173 "entity epsilon"
13. L239 - 241 this paragraph does not contribute to this section
14. L256 errant full stop
15. L326 "method" instead of AVG_T would convey this better
16. L404 Hits@K is never used in the paper
17. L420 reference the ablation study here and show it it supports your argument
18. L450 an example of one of the queries in the main text would be nice
19. L535 just one metric is defined, not multiple

**Questions:**

1. L30 in what sense is the new MRR metric validated by your results?
2. L105 what are the inherent fuzzy relationships within numerical data?
3. L214 why is this referred to pre-training? Is it not just "training"?
4. L236 what does it mean that "X represents various relationships"?
5. L247 what does "original distribution" refer to?
6. L264 what is an "anchored" entity?
7. L267 is N here referring to the set of all possible numbers in the system? This will then be a huge vector
8. L308 what are these "membership functions"?
9. L329 what do Avg_All and the other columns denote?
10. L408 are these values only eliminated post-training?
11. L465 if you’re using a continuous range, is this not the only way this can be done?
12. L524 how does your work differ from that of LitCQD?

---

### Official Review · Reviewer_F4o8 · 2024-10-30

**Soundness:** 2
**Presentation:** 2
**Contribution:** 2
**Rating:** 3
**Confidence:** 5

**Summary:**

The paper proposes an approach (CNR-NST) for complex query answering on incomplete knowledge graphs. In contrast to many previous works (e.g., CQD, GQE, ...), the paper supports knowledge graphs with numeric attributes. The proposed approach is compared to one of the existing works addressing the same problem (NRN).

**Strengths:**

- The paper tackles a highly relevant problem. Numeric data is wide-spread in real-world knowledge graphs.
- The experimental results show that the new approach outperforms the NRN baseline in most cases.

**Weaknesses:**

- Lack of novelty: The previous approach LitCQD is mostly ignored leading to wrong claims about the contribution (see details)
- Insufficient experiments: LitCQD is missing as a baseline
- Poor technical and presentation quality (see details below and questions)

### Details

- Figure 1: Q3: "What is the total population of Schleswig-Holstein and Dakar?" The paper claims that previous approaches could not answer this query ("cannot compute or infer new numerical answers from multiple values (like Q3)."). LitCQD supports to answer such queries (see Equation 13 in the LitCQD paper).

- "Numerical Binary Operation Operator." This operator can handle "queries that involve numerical answers" and this operator is presented as a novel contribution. However, the Section 4.2 "Multihop Queries with Literals and Literal Answers" in the LitCQD paper deals with exactly this issue.

- Section 4.3 : "For the first time, we extend numerical reasoning in knowledge graphs to the real number domain, whereas previous methods were confined to the discrete numerical domain within the KG." This sentence is wrong (c.f. LitCQD)

- Section 4.4: "Previous approaches used the same evaluation metrics for these queries as for entities, but this method has limitations." This sentence is wrong. LitCQD used Mean Absolute Error (MAE) and Mean Squared Error (MSE) instead of Mean Reciprocal Rank (MRR).

- Abstract: A sentence seems to be repeated in the abstract: "The proposed frame-work is the first to enable binary…" and "The CNR-NST framework can perform binary…". Apart from the fact that this sentence is wrong (see LitCQD), the two sentences should be merged.

- Preliminaries: After the period, there should always be a space. For example: "relations R.Each triplet" --> "relations R. Each triplet" (lines 150-151, to show only a few. The problem occurs more often throughout the paper.)

- Preliminaries "Knowledge Graph $G = (V, R, \epsilon)$ contains the set of all entities $\epsilon$ and the set of all relations $R$." What is $V$ in this definition? How is it different from $\epsilon$? This definition seems wrong: a knowledge graph is not only defined by its set of entities and relations. Triples define a knowledge graph, too.

- Preliminaries: Lines 173-174: "In the above equation, the variable $E$ represents a subset of entity $\epsilon$..." How is a variable a subset? It might be better to talk about variable bindings.

- Preliminaries: The functions $r_i$ and $a_j$ are mentioned in the paragraph on lines 173-178 but never defined

- Section 3.1: Confusing notations are used, e.g., it is not clear whether $\mathbb{R}$ is the usual real numbers, see Equation (4). Also see line 309 where $\mathcal{R}$ (instead of $\mathbb{R}$) is defined as the real number domain. Moreover, in line 152 $\mathcal{R}$ is defined as set of relations.

- Section 3.1: There seem to be many inconsistencies in the Methodology section. First it is written that $f(h,t,r) \in \mathbb{R}$ then $(h,t,r) \in \mathbb{R}\cup \mathbb{A} \cup \mathbb{F}$ (Equation 4). Both f(h, t, r) and (h, r, t) are in $\mathbb{R}$ (with and without the $f$)? Moreover, the notation $(h,r,t)$ is not consistent across the paper. Sometimes a triple is denoted as $(h,t,r)$.

- Section 3.2: Confusing terms are used on page 6: "relation edge" and "entity node". I believe one should either use "relation" or "edge" or "entity" or "node" but not two of the words.

- Section 3.2: $V^*(X = x)$ is defined as a truth value (see line 273). The transposition operation is also applied to it afterwards, see Equation (9). This makes the equation look incorrect. Important details on how fuzzy numbers and fuzzy sets are used in the proposed approach seem to be missing.

- Section 3.2: Lines 270-271: It is written that $\mathcal{U}(Q)$ denotes a probability, but it is actually defined as a set in Equation (8)

- Section 3.2: Lines 274-275: If $x$ is an entity or numerical value, what is $|x|$?

- Equation 10: It should be $\phi (V_{i1}, V_{i2}, \ldots, V_{im})$, you used $V_{i1}$ twice in the enumeration.

**Questions:**

- How does the proposed approach compare to LitCQD?
- Section 3.1: It seems the loss function in Equation (4) can be negative (the second part). How would one train with a negative loss? Are there any settings that prevent the loss from becoming negative?
- Lines 464-465: "Instead of ranking based on the exact match of numerical nodes, we compute the RANK using the probability ranking of numerical nodes whose relative error compared to the correct answer is below a specified threshold (typically set at 0.001)"
This evaluation metric discards some numerical nodes, which might not reveal the actual performance of a model. As an example, assume that $n$ nodes were ranked higher than the target node in the original metric (MRR). Also assume that these $n$ nodes are now removed in your new metric because they do not fulfil the 0.001 criterion. Then, the actual target node will now be ranked 1st, which does not really reveal the performance of your model. Any ideas on how to improve this metric? Why don't you use Mean Absolute Error (MAE) or Mean Squared Error (MSE) as LitCQD does?

---

### Official Review · Reviewer_NDES · 2024-11-01

**Soundness:** 2
**Presentation:** 3
**Contribution:** 2
**Rating:** 3
**Confidence:** 4

**Summary:**

This paper proposes a CQA method on KGs with numeric values and binary operations. This approach can effectively handles more than 100 types of complex numerical reasoning queries. On three public datasets, the proposed method CNR-NST demonstrates SOTA performance in complex numerical queries, achieving an average improvement of over 40% compared to existing methods.

**Strengths:**

1) The first work considering complex queries involving binary numeric operations.
2) Experimental improvements seem significant.

**Weaknesses:**

1) A comparision with some naive baselines could significantly improve the perception of the experimental results. Especially for the query types with binary operations. The MRR numbers are very small and there is no baseline, and hence it is very hard to judge whether the results are good or not. For example, one could use some simple numeric rules mined from the training graph to derive answers.
2) It seems that the techniqical contributions are two-fold: 1) Multi-ComplEx, which is a direct extension of ComplEx used in CQD to deal with numerical information; 2) The numerical computation framework. However,  it is unclear whether the numerical computation is a reasonable or not. Does it satisfy some laws like commutative, associative and distributive Laws? I see no discussion about this but I think this is the key which influences the generalization capability of the reasoning.
3) The test queries are generated as "hard queries" in the sense that as least one missing link is in the test graph. However, it is unclear for a multi-hop query, how much percent of the links are seen in the training graph. Note that this is important, as if most of the links in a multi-hop queries are seen. Then the problem can be reduced to a link prediction problem.

**Questions:**

1) Why LitCQD is mentioned but not compared?
2) Why equation 9 is used, does it satisfy commutative, associative and distributive laws, and many others?

---

### Official Review · Reviewer_rXou · 2024-11-04

**Soundness:** 4
**Presentation:** 3
**Contribution:** 4
**Rating:** 8
**Confidence:** 3

**Summary:**

The authors introduce a novel method for numerical reasoning over numeric knowledge graphs containing both real-valued continuous attributes and entities and relations. This is important as it allows for more robust reasoning over and modelling of real-world natural and common queries within KGs. One novelty of the method lies in the capability to encode the numeric and entity attributes separately, allowing for the use of binary operators to obtain numeric values outside of the designated knowledge graph. A more comprehensive evaluation suite is proposed for this type of OOD (meaning numbers/ents not in the graph) setup. The method allows the joint encoding (separately within a joint training process) of entity-numeric relationships that capture the semantics and abstractions of the relation. This is achieved using Multi-ComplEx, an extension of the link prediction method ComplEx (a strong method) by combining a separate set of numeric-value and numeric-entity encodings.

**Strengths:**

The research includes many merits, from the novel approach to tackle numeric reasoning within the KGs that is able to achieve ~40% increase compared to prior benchmarks, to the introduction of more robust testing/evaluation query types (2b, 3b ... etc). The study is supported by straightforward benchmarks and comprehensive evaluations, showing that the method is particularly well suited for numeric reasoning and is computationally advantageous as it does not require explicit training on complex queries (trained only on atomic queries), yet generalises well to complex reasoning structures. The use of Multi-ComplEx is shown to be essential for embedding the numeric information within the KG, while the use of fuzzy sets allows robust reasoning when dealing with direct numeric operations, comparisons and assessments.

**Weaknesses:**

1. While the method works well with a margin of error and allows the obtaining numeric values outside of the KG, it is still limited in terms of the numeric continuous values that it predicts and the precision of such numbers through the limited amount of used binary operators and initially present Numeric values. Can the framework solve absurd queries of type "Rope $X$ is $1$mm, Y is $1.61$Meters how long does rope $Z$ has to be to be 10^3 time longer than squared average of $X$ and $Y$"?

2. As the framework shares many similarities with CQD, a natural question arises if the intermediate answers obtained during query answering are calibrated to interact with each other (intermediate probability ranges are similar), which was a problem in CQD outlined in CQD-A. This is particularly important as the fuzzy aggregation method (the T-norm) that was chosen is the product norm, which suffers from this discrepancy.

**Questions:**

1. Context in Weaknesses Point 1: Can the framework solve absurd queries of type "Rope $X$ is $1$mm, Y is $1.61$Meters how long does rope $Z$ has to be to be 10^3 time longer than squared average of $X$ and $Y$"?
2. Context in Weaknesses Point 2: Does the method suffer from calibration issues (ranges of probabilities are not homogenous, do not interact) within separate intermediate answers similar to CQD, meaning that each intermediate top-k answer can fall within varying ranges of probability and be omitted/filtered out during product (the t-norm chosen) aggregation?

---

### Meta-Review · Area_Chair_jooR · 2024-12-18

**Metareview:**

The reviewers saw several positive things in the paper. However, the submitted version is the one that they have to evaluate on and it was felt that the mathematical sloppiness, and lack of clarity are some key issues and this paper was in true borderline. However, from the discussion, it is clearer that the paper needs a major round of edits (if this was a journal, this would be a major revision). Hence, the paper cannot be accepted in current form but the authors are encouraged to fix the paper and submit to another suitable venue.

**Additional Comments On Reviewer Discussion:**

There was some discussion among the reviewers and some of them directly engaged with the authors. It was clear that the paper had some merits and the reviewers were thankful to authors for running some more experiments and stuff, but it was suggested that the paper needs one more round of fixing before acceptance.

---

### Decision · Program_Chairs · 2025-01-22

Reject